

# Development and evaluation of a suite of isotope reference gases for methane in air

P. Sperlich[1, 2], N. A. M. Uitslag[1,3], J. M. Richter[1], M. Rothe[1], H. Geilmann[1], C. van der Veen[4], T. Röckmann[4], T. Blunier[5] and W. A. Brand[1]

[1]Max-Planck-Institute for Biogeochemistry (MPI-BGC), Jena, Germany

[2]National Institute of Water and Atmospheric Research (NIWA), Wellington, New Zealand

[3]Centre for Isotope Research (CIO), Energy and Sustainability Research Institute Groningen (ESRIG), University of Groningen, Groningen, The Netherlands

[4]Institute of Marine and Atmospheric Science in Utrecht (IMAU), Utrecht, The Netherlands

[5]Centre for Ice and Climate (CIC), Niels Bohr Institute, University of Copenhagen, Copenhagen, Denmark

*Correspondence to*: P. Sperlich (peter.sperlich@niwa.co.nz)

**Abstract.** Measurements made by multiple analytical facilities can only be comparable if they are related to a unifying and traceable reference. However, reference materials that fulfil these fundamental requirements are unavailable for the analysis of isotope ratios in atmospheric methane, which led to misinterpretations of combined data sets in the past. We developed a method to produce a suite of standard gases that can be used to unify methane isotope ratio measurements of laboratories in the atmospheric monitoring community. We calibrated a suite of pure methane gases of different methanogenic origin against

international referencing materials that define the VSMOW and VPDB isotope scales. The isotope ratios of our pure methane gases range between −320 and +40 ‰ for $\delta^2$H-CH$_4$ and between −70 and −40 ‰ for $\delta^{13}$C-CH$_4$, enveloping the isotope ratios of tropospheric methane (about −90 ‰ and −47 ‰ for $\delta^2$H-CH$_4$ and $\delta^{13}$C-CH$_4$, respectively). We estimate combined uncertainties for our $\delta^2$H and $\delta^{13}$C calibrations of <1.5 ‰ and <0.2 ‰, respectively.

Aliquots of the calibrated pure methane gases have been diluted with methane-free air to atmospheric methane levels and filled

into 5-L glass flasks. These synthetic gas mixtures comprise atmospheric oxygen/nitrogen ratios as well as appropriate argon, krypton and nitrous oxide mole fractions to prevent gas-specific measurement artefacts. The resulting synthetic atmospheric reference gases will be available to the atmospheric monitoring community. This will provide unifying isotope scale anchors for isotope ratio measurements of atmospheric methane so that data sets can be merged into a consistent global data frame.




## 1 Introduction

Isotope ratios of trace gases in the present and the past atmosphere (e.g. from ice cores) are an emerging tool to study the biogeochemical processes that cause the variation of $CH_4$ in the atmosphere (Stevens and Rust, (1982), Lowe et al., (1994), Sapart et al., (2012), Sperlich et al., (2015)). Recently, two conflicting publications highlighted *i)* the interpretative power

when data sets from multiple laboratories are combined for spatio-temporal analysis of $CH_4$ isotope ratios (Kai et al., 2011) and *ii)* the pitfalls when differences due to laboratory offsets are mis-interpreted as spatial variability of $CH_4$ sources (Levin et al., 2012). Levin et al., (2012) identified calibration offsets between three laboratories by comparing their long term observations in Antarctica where the carbon isotopic composition of $CH_4$ is assumed to be free of spatial gradients. However, this technique is a temporary work-around that excludes the use of data sets from laboratories without a history of observations

in Antarctica or a traceable link to Antarctic observations. If a unique set of reference material were available to all laboratories that measure isotope ratios of atmospheric $CH_4$, the science community would be able to combine the available measured data sets to investigate spatio-temporal variations. However, such a sustainable solution that unifies the measurements of all laboratories has not been established.

The lack of unique reference gases has long been recognised in the literature on $CH_4$ isotope ratios, ranging from pioneering

papers (e.g. Craig (1953), Schiegl and Vogel, (1970)) to recent technical publications on isotope ratios in atmospheric $CH_4$ (e.g. Sperlich et al., (2013), Tokida et al., (2014), Eyer et al., (2015)) as well as papers that present and interpret such data (Levin et al., 2012). So far, laboratories have either developed methods to calibrate purified $CH_4$ against international reference material of different physico-chemical properties (e.g. Schiegl and Vogel, (1970), Stevens and Rust, (1982), Dumke et al., (1989), Levin et al., (1993), Lowe et al., (1994), Quay et al., (1999), Sperlich et al., (2012)) or they reference their $CH_4$ samples

to standards that were propagated from calibrations elsewhere (e.g. Bergamaschi et al., (1994), Brass and Röckmann, (2010) , Bock et al., (2014), Schmitt et al., (2014), Rella et al., (2015), W.A. Brand 2015 pers. comm.). The diversity of referencing trajectories can lead to significant referencing offsets between the laboratories as shown by Levin et al., (2012).

Ghosh et al., (2005) established a method to harmonise isotope ratio measurements of atmospheric $CO_2$. Based on this method, the ISOLAB of the Max-Planck-Institute for Biogeochemistry (MPI-BGC) in Jena, Germany, distributes a suite of reference

gases, known as the "JRAS air set", which is accepted as an isotope scale anchor by the community (WMO, 2012). Calibrating against the "JRAS air set" reduces laboratory offsets and helps reaching and maintaining the compatibility goal (Wendeberg et al., 2013).

This paper describes a method to produce isotope reference gases for atmospheric $CH_4$. The reference gases we produced bracket the isotopic ratios of tropospheric $CH_4$ and span over a large isotopic abundance range, which enables a two-point

calibration to account for scale compression effects, thereby reducing the referencing uncertainty (Coplen et al., 2006). These isotope reference gases are produced from a well calibrated suite of $CH_4$ gases and will be made available from MPI-BGC. Our reference gases for $CH_4$ isotope ratios may help the community to reach the compatibility goals of 1 ‰ and 0.02 ‰ for $\delta^2$H-$CH_4$ and $\delta^{13}$C-$CH_4$, respectively (WMO, 2014).



## 2 Materials and Methods

The aim of this paper is to present the method that we use to calibrate and prepare isotope reference gases for $CH_4$ in air samples, as outlined in the flow diagram of Fig. 1. Our method is based on the initial calibration of two pure Master-$CH_4$ gases for their $\delta^2H$-$CH_4$ and $\delta^{13}C$-$CH_4$ isotope ratios against international isotope reference materials. The two Master-$CH_4$ gases are then used to calibrate a suite of further $CH_4$ gases. Once calibrated, aliquots of the pure $CH_4$ gases are diluted with $CH_4$-free air to atmospheric $CH_4$ mixing ratios. The resulting synthetic air standard can be distributed and analysed in a similar fashion as atmospheric samples by collaborating laboratories, following the Principle of Identical Treatment, PIT, (Werner and Brand, 2001).

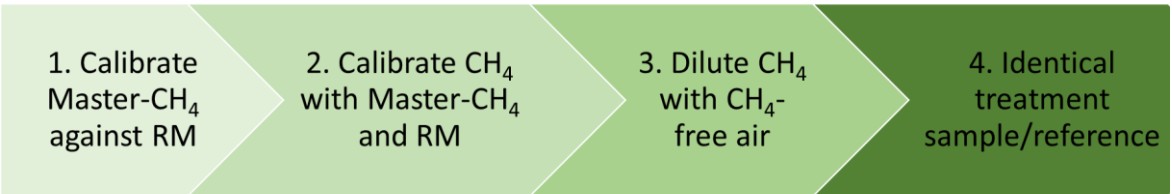

Figure 1: Flow diagram of methodological steps to produce synthetic isotope reference gases for $CH_4$ in air. RM is the abbreviation for established international Reference Materials.

### 2.1 Gases and reference materials used in this study

Our study is based on a suite of $CH_4$ gases that differ in their commercial provider, methanogenic origin and isotopic composition. We identify our $CH_4$ gases by names as shown in Table 1. "Biogenic" and "Fossil" have been calibrated in a previous $CH_4$ referencing study (Sperlich et al., 2012) and are therefore of known isotopic composition. Treating these gases as unknowns in this study allows testing and evaluating the performance of the presented methods. Six other $CH_4$ gases were purchased from suppliers of commercial gases or laboratory equipment (Air-Liquide, Westfalen AG, Linde, Messer, Campro Scientific) and were used as purchased or as mixtures thereof. The purity level of all our $CH_4$ gases is 99.995% or higher.



Table 1: Gases used for this study. Note that Mike-1 and Martha-1 were created as gas mixtures but were topped up with further gases to create Mike-2 and Martha-2, respectively. Thus, Mike-1 and Martha-1 do not exist anymore.

| Gas name | Cylinder volume [L] | pressure [bar] | function in study | $CH_4$ source | gas supplier |
|---|---|---|---|---|---|
| Megan | 10 | lost | 1st Master $CH_4$ | fossil $CH_4$ | Air Liquide |
| Merlin | 10 | 190 | 2nd Master $CH_4$ | fossil $CH_4$ | Air Liquide |
| Mike-1 | - | - | calibration $CH_4$ | MPI mixture | |
| Mike-2 | 5 | 45 | calibration $CH_4$ | MPI mixture | |
| Merkur | 2 | 100 | calibration $CH_4$ | fossil $CH_4$ | Messer Griesheim |
| Merida | 10 | 175 | calibration $CH_4$ | Unknown | Westfalen AG |
| Martha-1 | - | - | calibration $CH_4$ | MPI mixture | |
| Martha-2 | 10 | 165 | calibration $CH_4$ | MPI mixture | |
| Minion | 3 | 150 | calibration $CH_4$ | Unknown | Messer Griesheim |
| Melly | 50 | 193 | calibration $CH_4$ | Unknown | Westfalen AG |
| $\delta^2H$-Spike gas | 0.4 | 2.5 | spiking gas $CH_3D$ | | Campro Scientific |
| Fossil | 30 | 2 | calibration control | fossil $CH_4$ | Air Liquide Denmark |
| Biogenic | 30 | 2 | calibration control | biogas plant | Biogas Plant |
| synthetic air | 50 | 200 | synthetic air matrix | | Linde |
| Krypton | 2 | 200 | synthetic air matrix | | Westfalen AG |
| Carina-1 | 50 | 200 | isotope scale | Jena air | MPI-BGC |
| Carina-2 | 50 | 200 | isotope scale | Jena air | MPI-BGC |

"Megan" and "Merlin" were used as Master-$CH_4$ gases that were calibrated against international reference materials for their hydrogen and carbon isotopic composition. Table 2 summarises information on the applied reference materials.

Table 2: Reference materials used in this study. Column 1 and 2 show name and chemical formula, respectively. "RM" in column 3 indicates an international reference material while "ws" identifies local working standards. The $\delta^2H$ and $\delta^{13}C$ isotope ratios are shown in column 4 and 5, respectively, while column 6 and 7 list the sources and related publications of reference materials we used. We show the most recent values of reference materials as summarized by Brand et al., (2014). For MPI-BGC data the uncertainties correspond to the 95 % confidence limit of the error of the mean, i.e. the standard error of the mean, multiplied by Student's t-factor. The * indicates an uncertainty estimate that is currently under investigation as explained in the main text.





| Name | Material | int/ws | $\delta^2$H [‰] | $\delta^{13}$C [‰] | Source | Reference |
|------|----------|--------|------------|-------------|--------|-----------|
| VSMOW-2 | $H_2O$ | RM | $0 \pm 0.3$ | -- | IAEA | Gröning et al., (2007) |
| SLAP-2 | $H_2O$ | RM | $-427.5 \pm 0.3$ | -- | IAEA | Gröning et al., (2007) |
| GISP | $H_2O$ | RM | $-189.7 \pm 0.9$ | | IAEA | Brand et al., (2014) |
| LSVEC | $Li_2CO_3$ | RM | -- | $-46.6 \pm 0.03*$ | IAEA | Coplen et al., (2006), |
| CO-9 | $BaCO_3$ | RM | -- | $-47.32 \pm 0.06$ | IAEA | Coplen et al., (2006) |
| www-j1 | $H_2O$ | ws | $-67.0 \pm 0.4$ | | MPI-BGC | -- |
| BGP-j1 | $H_2O$ | ws | $-187.1 \pm 0.6$ | | MPI-BGC | -- |
| Mar-j1 | $CaCO_3$ | ws | | $1.96 \pm 0.01$ | MPI-BGC | Brand et al., (2009b) |
| ali-j3 | Acetanilide | ws | | $-30.06 \pm 0.05$ | MPI-BGC | -- |

## 2.2 Referencing CH₄ for $\delta^2$H against VSMOW/SLAP and against other CH₄ gases

We use a high temperature conversion elemental analyser (TC/EA) coupled to an IRMS (Delta Plus XL, Thermo Finnigan,

Bremen, Germany) via an open split (ConFlo III, Thermo Finnigan, Bremen, Germany) to convert $CH_4$ to $H_2$ (+ carbon) and measure $\delta^2$H-$CH_4$ in pure $CH_4$ gases after the conversion to $H_2$ (Werner et al., 1999). This system is routinely used for the analysis of hydrogen and oxygen isotope ratios in water samples that are injected through a heated septum at (130°C) into a glassy carbon reactor where the $H_2O$ is converted to CO and $H_2$ at temperatures above 1400°C (Gehre et al., (2004), Hilkert et al., (1999)). In order to reference the hydrogen isotopic composition of pure $CH_4$ against international reference waters

(VSMOW-SLAP scale), we inject pure $CH_4$ into the TC/EA using a two position 10-port valve (VICI, USA) that is configured as shown in Fig. 2. The outlet flow of this valve is routed through the septum also used for water injection so that the helium gas stream of 15 mL/min carries the $CH_4$ from the 1 mL sample loops into the TC/EA reactor. Typical $CH_4$ feed flow rates range between 2-3 mL/minute. The upstream plumbing can be configured so that every valve switch injects an aliquot of the same or of two alternating $CH_4$ gases into the reactor. Measurement sequences can now be created by injecting $CH_4$ gases

manually and reference waters automatically (via injections through the septum from an autosampler) into the reactor, where both $CH_4$ and $H_2O$ are converted to molecular hydrogen. Apart from the injection procedure itself, the generated $H_2$ of both $CH_4$ and international reference waters conforms with the Principle of Identical analytical Treatment (PIT) (Werner and Brand, 2001). The amounts of injected $CH_4$ and $H_2O$ are adjusted in our experiments to achieve matching peak shapes and amplitudes during IRMS analysis in order to minimize $H_3$-factor variations between $CH_4$ and $H_2O$ samples. The resulting peaks for $CH_4$

samples and reference waters are shown in Fig. 4. Following this method, we calibrated "Megan" and "Merlin" (Table 1) in three independent repetitions during three days against the in house standards "www-j1" and "BGP-j1", which were independently calibrated against international reference waters VSMOW2 and SLAP2 (Table 2).




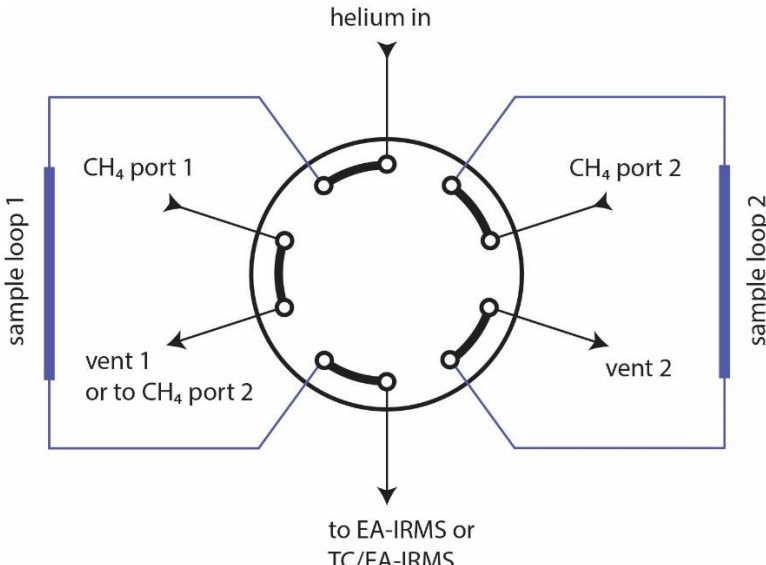

Figure 2: Configuration of two-position 10-port valve with plumbing for helium carrier gas, two ports to introduce the $CH_4$ sample and two 1-mL sample loops as indicated by thick blue lines. Note that vent 1 can be connected to $CH_4$ port 2 in order to introduce the same $CH_4$ gas with every injection. Sample loop 1 is vented through vent 1 for the injection of two alternating $CH_4$ gases.

The two sample loops attached to the 10-port valve (Fig. 2) can be fed by two different $CH_4$ gases. This configuration enables alternating measurements of unknown $CH_4$ against known $CH_4$ in an identical fashion. When used in this configuration, the previously calibrated "Megan" and "Merlin" served as known reference gases to calibrate the other $CH_4$ gases (Table 1) to the respective isotope scales. Additional injections of $H_2O$ standards were used to control isotope scale contraction during $\delta^2H$-$CH_4$ analysis.

Because the $\delta^2H$-$CH_4$ range in our samples is very large (~350 ‰, Table 3), sufficient control of potential scale compression effects is imperative (Renpenning et al., 2015). We achieve this by applying reference waters that span the full VSMOW2 to SLAP2 isotopic range from 0 to −427.5 ‰. The daily working reference waters (www-J1 and BGP-J1) cover a $\delta^2H$ range of −67.0 to −187.1 ‰ (Table 2). Our suite of reference waters enables a multi-point calibration that largely envelopes the $\delta^2H$ range of the $CH_4$ samples (Table 2 and 3).

In most routine isotope ratio measurements, samples are referenced against a standard of identical composition, following the principle of identical treatment (Werner and Brand, 2001). For example, measurement artefacts cancel when water samples are calibrated against reference waters. However, great care has to be taken when comparing two different materials, as is the case when calibrating $CH_4$ against reference waters. This is because standard and sample require material-specific attention to prevent isotope fractionation during analysis. Calibration errors may occur when *i)* only one of the two materials or *ii)* both



materials are incompletely transported or converted, or when memory effects inside the sample conversion system are different for the respective materials. In order to rule out calibration artefacts due to the analysis of two chemically very different materials and in order to ensure that our method is suitable for quantitative analysis of both $CH_4$ and $H_2O$ without isotope fractionation, we performed several series of experiments to optimise the reactor and septum temperatures, the compound-

specific memory effects and the applied sample size as described in detail in Appendix 1.

The introduction of $H_2$ samples into the ion source of an IRMS leads to the formation of $H_3^+$ ions that are registered on the $HD^+$ detector, which is accounted for by the so called "$H_3$-factor correction" (Friedman, (1953), Sessions et al., (2001)). While the $H_3$-factor correction is experimentally determined and kept constant throughout a series of measurements, the $H_3^+$ formation may be dynamic over time. In order to minimise the impact of inaccurate $H_3$-correction between $CH_4$ and $H_2O$ analysis, we

matched the $H_2$ peak heights resulting from both $CH_4$ and $H_2O$ injections around $5.5 \pm 0.5$ V. Typical peak widths were around 45 s and 60 s for $H_2O$- and $CH_4$-derived $H_2$ peaks, respectively. A typical chromatogram of the $\delta^2H$-$CH_4$ calibration including details on peak shape and background is shown in Fig. 3. This figure shows that both the $CH_4$-derived and the $H_2O$-derived $H_2$ peaks are of comparable height and width, which justifies the analysis of sample and reference material using the standard integration software (ISODAT, Thermo). We performed a range of experiments to test for systematic analytical effects that

could impact on the isotope analysis of $CH_4$ and/or $H_2O$ and used the results of these experiments to optimize the analysis. These experiments are described in detail in Appendix 1. The good control of peak shapes and the elimination of analytical artefacts allow for the conclusion that our method is suitable to calibrate $CH_4$ against reference $H_2O$ with high precision and accuracy.





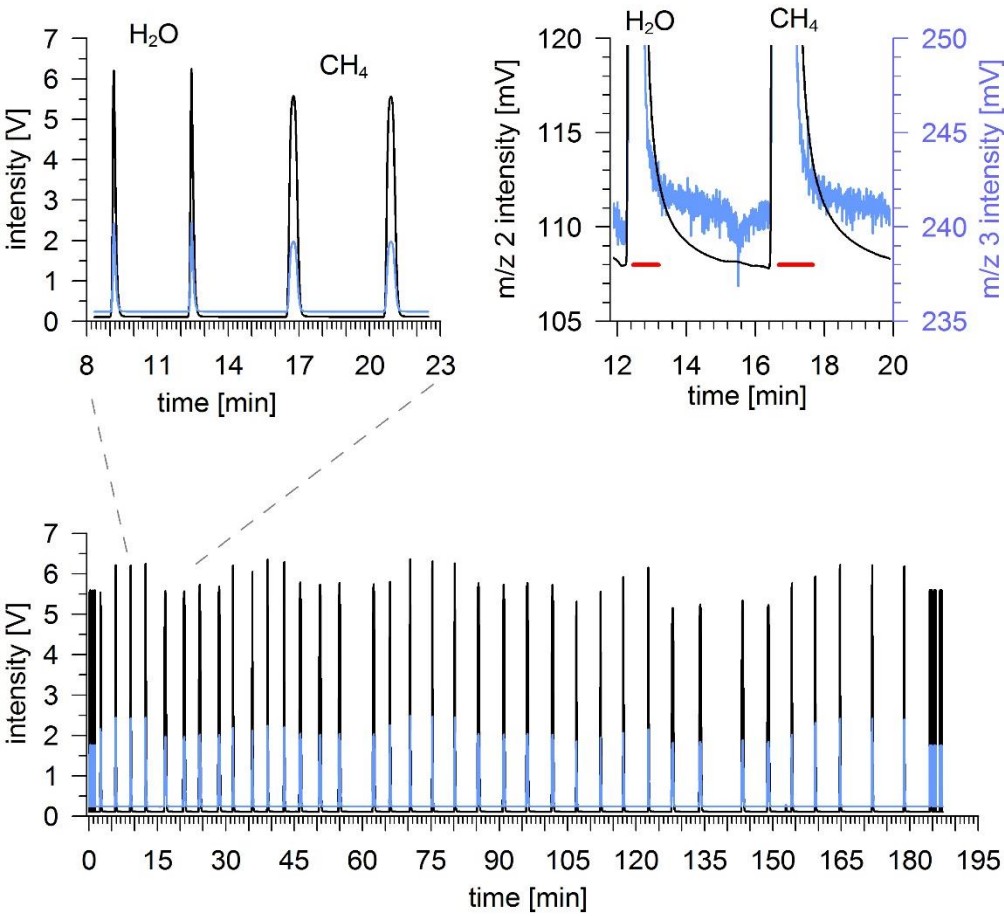

Figure 3: Chromatograms of $\delta^2$H-CH$_4$ calibration sequences using TC/EA-IRMS with traces of *m/z* 2 and *m/z* 3 shown in black and blue, respectively. The bottom panel shows an example of an entire calibration sequence which begins with 3 square-shaped peaks of pure H$_2$, followed by alternations of 3 to 4 H$_2$O- and 3 to 4 CH$_4$-derived H$_2$ peaks before the sequence ends with another 3 square shaped peaks of pure H$_2$. The top left panel enlarges 4 peaks that are derived from H$_2$O (peak # 6-7) and CH$_4$ (peak # 8-9), respectively. A zoom into baseline details of H$_2$O-derived peak # 7 and CH$_4$-derived peak # 8 is shown in the top right panel. Red lines indicate the sections used for peak integration (weak widths are 43 s and 59 s for H$_2$O- and CH$_4$-derived H$_2$ peaks) by the IRMS software.





### 2.3 Referencing CH₄ against LSVEC / NBS 19

We calibrated $\delta^{13}C$-$CH_4$ in pure $CH_4$ gases after conversion to $CO_2$ using an elemental analyser (EA 1100, CE, Rodano, Italy) coupled to an IRMS (Delta Plus, Thermo Finnigan, Bremen, Germany) through an open split (ConFlo III, Thermo Finnigan, Bremen, Germany). This system is routinely used for the analysis of $^{13}C$ and $^{15}N$ in samples with solid or liquid matrices (Werner et al., (1999), Brooks et al., (2003)). For the $CH_4$ calibrations, we used the same 10-port valve assembly as described above (Fig. 2). We modified the EA autosampler to inject the $CH_4$ gas with a helium carrier gas stream of 10 mL/min through an extra 1/16" tube of 70/30 % Cu/Ni alloy into the oxygen plume region inside the combustion chamber of the EA (Fig. 4).

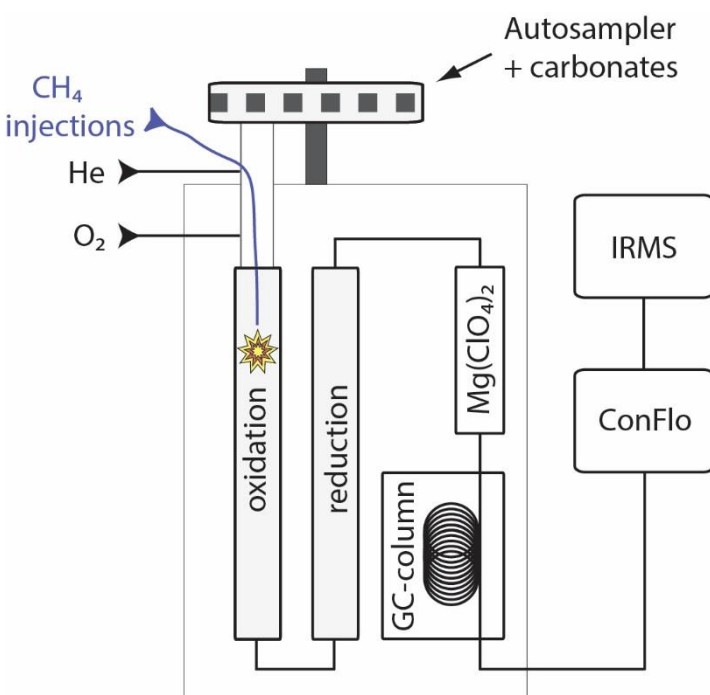

Figure 4: EA-IRMS system with additional inlet for gas injection into the combustion unit.

The plumbing of the system is designed so that gaseous and solid samples can be supplied to the same location inside the combustion reactor of the EA. All samples are oxidised at 1020°C (Werner et al., 1999) and experience identical analytical treatment after the combustion. After oxidation, the sample passes through a reduction reactor filled with elemental copper at 650°C to remove excess $O_2$ and to reduce $NO_x$ if present. The sample is dried using a combination of a Nafion™ dryer (Perma Pure LLC, Toms River, NJ, USA, not shown in Fig 4) and $Mg(ClO_4)_2$ before it is routed through a GC column that is held at 80°C (3m, 1/4", Porapak PQS, CE instruments). The sample is then transferred into the IRMS through an open split.





Alternating $CH_4$ injections and applications of reference materials such as LSVEC, Mar-j1 and ali-j3 (Table 2) via an autosampler enable direct referencing of $CH_4$ gas to the VPDB isotope scale. We used this method to calibrate "Megan" and "Merlin" as Master-$CH_4$ gases (Table 1) over three independent measurement periods each. Alternating injections of a Master-$CH_4$ and unknown $CH_4$ gases were then used to calibrate all other $CH_4$ gases (Table 1) for $\delta^{13}C$. Two chromatograms resulting

5  from both $CH_4$ and carbonate analysis using EA-IRMS are displayed in Fig. 5 and show very similar peak shapes for the analysis of both materials. Typical $m/z$ 44 amplitudes and peak widths were ~$7.4 \pm 0.2$ V and $101 \pm 1$ s for both materials, respectively.

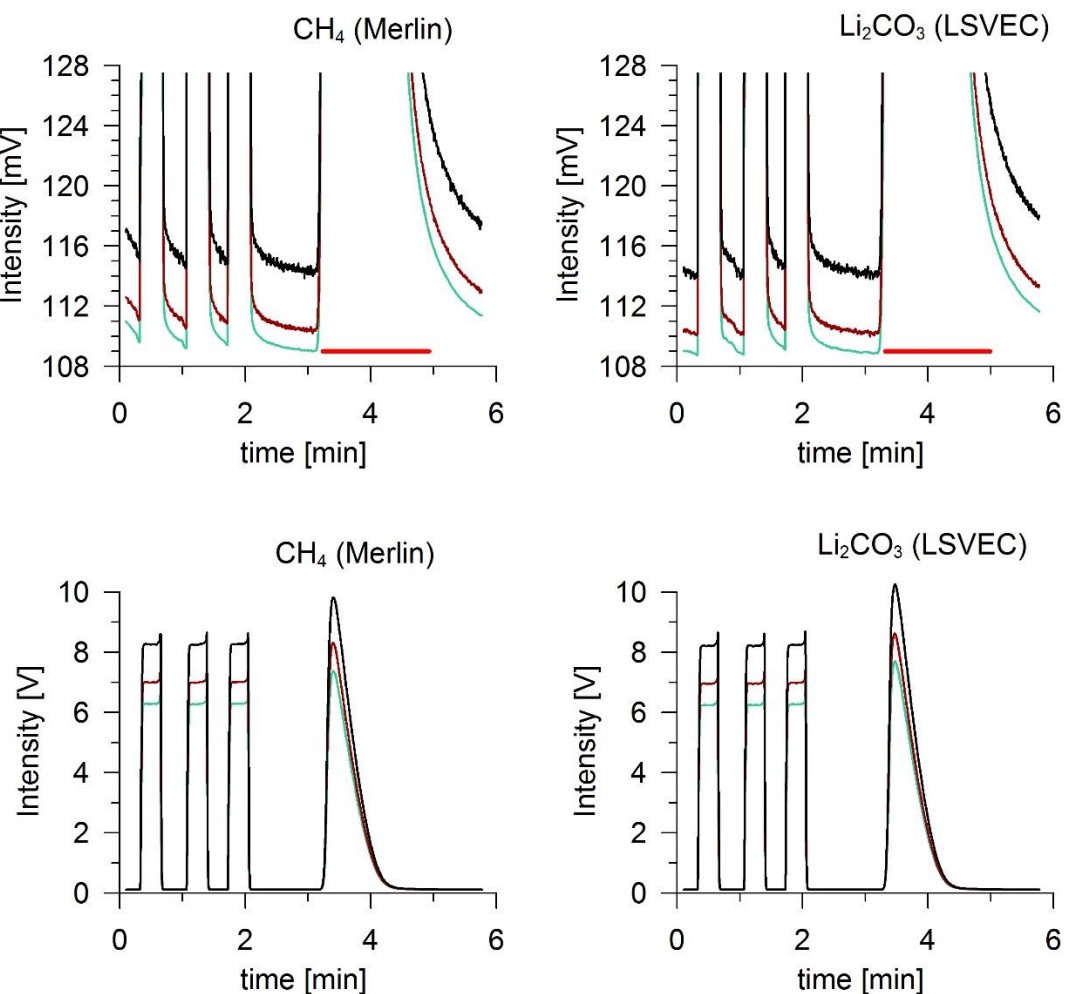

10  Figure 5: Chromatograms of $\delta^{13}C$-$CH_4$ calibrations using EA-IRMS with traces for $m/z$ 44, 45 and 46 represented by green, brown and black lines, respectively. Analysis of $CH_4$ (Merlin) and $Li_2CO_3$ (LSVEC) are shown in the left and right-hand panels, respectively. Bottom panels show the entire chromatogram while the two top panels zoom into the baseline of the traces. The first three square-shaped $CO_2$ peaks stem from injections of pure $CO_2$ working gas while the tailing peaks result





from $CH_4$- and $Li_2CO_3$-derived $CO_2$ analysis. The two red lines in the top panel indicate the sections that the IRMS software uses for peak integration (101 s and 100 s for $CH_4$- and $Li_2CO_3$-derived $CO_2$ peaks, respectively).

**2.4 Measurement uncertainty and error propagation**

With the two-step calibration strategy approach the error treatment also has two major components. For the $\delta^{13}C_{VPDB}$ calibration of the two Master-$CH_4$ gases Megan and Merlin we used the international reference materials NBS 19 and LSVEC with assigned $\delta^{13}C_{VPDB}$ values +1.95 and −46.6 ‰, respectively (Coplen et al., (2006), Brand et al., (2014)). These were analysed

within the same analytical sequences as the Master-$CH_4$ gases and used to position the $\delta^{13}C$-$CH_4$ values onto the international VPDB scale within every sequence by relating the $\delta^{13}C$-$CH_4$ value of each $CH_4$ peak to the known $\delta^{13}C$ values of the reference materials. The final uncertainty for the Master-$CH_4$ gases is given as the standard error of the mean of each sequence, multiplied by t, Student's factor for a 95 % confidence limit.

More recently, a variability between different LSVEC aliquots at the IAEA premises has been found which has been attributed

to interaction with atmospheric $CO_2$ and water, leading to a small increase in $\delta^{13}C_{VPDB}$ (S. Assonov, pers. comm.). The final $\delta^{13}C_{VPDB}$ value assigned to the specimen used in our laboratory might therefore be subject to a (small) adjustment, which will directly affect the calibrated methane values. We have tentatively made allowance for the effect by including an estimated uncertainty of ±0.03 ‰ for the LSVEC specimen used in the BGC ISOLAB experiments (Table 2). A final adjustment can only be made when a suitable solution to this issue has been found, agreed upon and published.

Similar to the $\delta^{13}C_{VPDB}$ calibration, the methane hydrogen isotopes have been calibrated using the international reference water samples VSMOW2 and SLAP2 with $\delta^2H_{VSMOW}$ of 0.0 and −427.5 ‰, respectively. Both of these reference materials have small uncertainties of ±0.3 ‰ for $\delta^2H_{VSMOW}$ (IAEA, (2009), Brand et al., (2014)). The uncertainties are included in the error values for the Master-$CH_4$ gases Megan and Merlin, which are otherwise obtained in an analogues fashion as described above: $CH_4$ and $H_2O$ sample injections are made within the same sequence. The observed water $\delta^2H$ values are then used to fix the

$\delta^2H$-$CH_4$ results to the VSMOW/SLAP scale. Again, the final $\delta^2H$-$CH_4$ values for the Master-$CH_4$ gases are presented as the standard error of the sequence means, multiplied by t, Student's factor for a 95 % confidence limit.

The secondary or "calibration" $CH_4$ gases have been directly analysed against the Master-$CH_4$ gases using identical procedures and equipment, switching the 10-port valve depicted in Fig. 2. The final errors evaluated for the gas-to-gas comparisons are obtained by the standard error of the calculated mean delta values, multiplied again by t, Student's factor for a 95 % confidence

interval. The uncertainty of the respective master methane gas is included in the combined error using standard error propagation. Possible scale contraction effects were treated routinely by measuring against a pair of reference water samples and applying a scale adjustment. This effect is part of the measurement uncertainty.





## 2.5 Producing synthetic isotope reference standards from pure CH₄ and CH₄-free air

The ISOLAB at the Max-Planck-Institute for Biogeochemistry operates a system to dilute pure $CO_2$ with $CO_2$-free air to atmospheric $CO_2$ mole fraction without isotopic fractionation (Ghosh et al., 2005). This system is referred to as ARAMIS and is used in this study to prepare atmospheric reference gases for $CH_4$. We dilute an aliquot of calibrated $CH_4$ with $CH_4$-free air

to atmospheric $CH_4$ mole fractions (~2 ppm) in 5-L glass flasks with a final filling pressure of 1.8 bar absolute. The $CH_4$-free matrix air has been target-mixed from ultra-pure constituents. It contains $N_2$, $O_2$, $N_2O$ and Kr at atmospheric mixing ratios. Krypton was added to this matrix air to account for the measurement artefact during GC-IRMS analysis of $CH_4$ for $\delta^{13}C$ (Schmitt et al., 2013), so that the produced isotope reference gases reflect a natural air sample as closely as possible. A sensitive analysis using high-precision gas-chromatography yielded an upper limit for the $CH_4$ mixing ratio of 0.5 ppb. Further details

on the matrix air and the krypton added are provided in Table 1. The $\delta^2H$ and $\delta^{13}C$ isotope ratios of $CH_4$ in the reference gases produced in the dilution experiments are measured with an average standard deviation (1$\sigma$) of 1.2 ‰ and 0.14 ‰, respectively, which is similar to the precision of the analytical system (1.0 ‰ and 0.12 ‰ for $\delta^2H$-$CH_4$ and $\delta^{13}C$-$CH_4$, respectively (W.A. Brand, 2015, pers. comm.) and suggests reproducible dilution of $CH_4$.

## 2.6 Comparison of isotope scales at MPI-BGI and IMAU

A new system to measure both carbon and hydrogen isotope ratios in atmospheric methane has recently been built at MPI-BGC, (W.A. Brand, pers. comm.). This system is referred to as *i*SAAC, in abbreviation for *i*sotope System for Analysing Atmospheric Constituents. *i*SAAC has been operational since 2012 with an external precision of 1.0 ‰ and 0.12 ‰ for $\delta^2H$-$CH_4$ and $\delta^{13}C$-$CH_4$, respectively, as determined using the performance chart technique (Werner and Brand, 2001) over a period

of 3 years. We calibrated our measurements using atmospheric working standards referred to as "Carina-1" and "Carina-2" (Table 1). The "Carina" gases have been calibrated at the Institute for Marine and Atmospheric Research in Utrecht (IMAU), using the analytical setup described by Brass and Röckmann, (2010) and Sapart et al., (2011). The history of the isotope scale at IMAU is described in detail by Brass and Röckmann (2010). We arbitrarily chose "Carina-1" as master reference gas for the *i*SAAC system. While "Carina-1" and "Carina-2" agree well in $\delta^{13}C$-$CH_4$, their previously calibrated $\delta^2H$-$CH_4$ value differs

by ~4 ‰. This offset has been confirmed as an artefact of the calibration at IMAU.

The synthetic isotope reference gases that were produced from previously calibrated $CH_4$ gases were analysed on *i*SAAC against "Carina-1". The results of these measurements are compared to the calibration results of the pure $CH_4$ gases so that the differences between the calibrated isotope ratios and the measurements against "Carina-1" indicate the offset between the new and the previous isotope scales in Jena (Sect. 3.3).




## 3 Results

### 3.1 Results for Master-CH$_4$ gas calibrations on the international VSMOW and VPDB isotope scales

Our Master-CH$_4$ gases "Megan" and "Merlin" have been calibrated for their carbon and hydrogen isotope ratios against international reference material and in house working standards (Table 2). The results are shown in Table 3. "Megan" was used as Master-CH$_4$ gas for all initial experiments and calibrations until it was accidentally vented to ambient in May 2015. In order to compensate the loss, "Merlin" was lifted in reference hierarchy from calibration CH$_4$ gas to Master-CH$_4$ gas as successor of "Megan". Therefore, Merlin was calibrated using two different methods, once against primary reference materials and once against a Master-CH$_4$ gas. Therefore, "Merlin" is listed twice in Table 3 with results that agree within the combined uncertainty.

Both "Megan" and "Merlin" are fossil CH$_4$ gases with typical dual isotope ratios for this source category (e.g. Quay et al., (1999), Mikaloff Fletcher et al., (2004)). The two Master-CH$_4$ gases are similar in $\delta^2$H-CH$_4$ with isotope ratios of −168.0 ± 0.6 ‰ and −165.7 ± 0.6 ‰, respectively. The calibrated $\delta^{13}$C-CH$_4$ isotope ratios of "Megan" and "Merlin" are −40.75 ± 0.07 ‰ and −39.07 ± 0.07 ‰, respectively.



Table 3: Results of CH$_4$ isotope calibrations. Each CH$_4$ gas is named in column 1 while its function is defined in column 2. Calibrated $\delta^2$H-CH$_4$ and $\delta^{13}$C-CH$_4$ values including combined uncertainties are listed in column 3 and 4, respectively. Note that results for "Merlin" are listed twice, once from its calibration against primary reference materials and once from calibration against Megan, the 1st calibrated Master-CH$_4$ gas. "Fossil" and "Biogenic" have previously been calibrated elsewhere (Sperlich et al., 2012). The results of the previous calibration are marked with (*) and are compared to the results of this method as discussed in the main text. Martha-1 and Mike-1 were intermittent gases used in the calibration comparison (Table 4) but were then mixed to be similar or more enriched in $\delta^2$H than tropospheric CH$_4$.

| Gas name | Function | $\delta^2$H-CH$_4$ [‰] | $\delta^{13}$C-CH$_4$ [‰] |
|---|---|---|---|
| Megan | Master | −168.1 ± 0.6 | −40.75 ± 0.07 |
| Merlin | Master | −165.7 ± 0.6 | −39.07 ± 0.07 |
| Merlin | Calibration | −164.1 ± 1.2 | −39.14 ± 0.19 |
| Martha-1 | Calibration | −176.6 ± 1.3 | −48.79 ± 0.06 |
| Martha-2 | Calibration | +36.2 ± 1.2 | −48.87 ± 0.15 |
| Mike-1 | Calibration | +44.5 ± 1.1 | −40.79 ± 0.08 |
| Mike-2 | Calibration | −80.3 ± 0.7 | −42.74 ± 0.10 |
| Merida | Calibration | −171.7 ± 1.1 | −60.28 ± 0.13 |
| Fossil | Calibration | −171.9 ± 1.6 | −39.72 ± 0.18 |
| Fossil | Comparison | −170.1 ± 0.7 (*) | −39.56 ± 0.04 (*) |
| Melly | Calibration | −177.5 ± 1.4 | −69.87± 0.11 |
| Minion | Calibration | −182.7 ± 1.2 | −58.06 ± 0.11 |
| Merkur | Calibration | −195.8 ± 1.2 | −43.03 ± 0.10 |
| Biogenic | Calibration | −319.8 ± 1.4 | −56.55 ± 0.11 |
| Biogenic | Comparison | −317.4 ± 0.7 (*) | −56.37 ± 0.04 (*) |

**3.2 Results for CH$_4$ gas calibrations against Master-CH$_4$ gases**

The calibration results of all secondary CH$_4$ gases are shown in Table 3. Our gases include gases from natural and commercial gas sources as well as isotopically spiked CH$_4$ mixtures. The gases cover large ranges in carbon (−70 to −39 ‰) and hydrogen isotope ratios (−320 to +36 ‰), which include the isotopic composition of tropospheric CH$_4$. For $\delta^2$H-CH$_4$ this was achieved by spiking fossil CH$_4$ gases with pure CH$_3$D to yield "Martha-2" and "Mike-1". In order to create a CH$_4$ gas with tropospheric $\delta^2$H-CH$_4$, "Mike-1" was diluted with a fossil CH$_4$ gas to produce "Mike-2", thereby using up "Mike-1".





### 3.3 Results for isotopic measurements of $CH_4$ in synthetic air standards

Aliquots of the previously calibrated $CH_4$ gases were diluted with $CH_4$-free air (Sect. 2.4) for analysis on a setup named "$i$SAAC" that has recently been developed at MPI-BGC for the analysis of dual isotope ratios of $CH_4$ in air samples (Sect. 2.5). In $i$SAAC, the diluted $CH_4$ reference gases are treated as unknown samples and referenced against "Carina-1". The

difference between the calibration results of the pure $CH_4$ gases and the diluted $CH_4$ gases is calculated by $\delta_{i\text{SAAC}} - \delta_{\text{pure}}$, which shows the calibration difference between the two different isotope reference materials. Table 4 displays the observed differences between the two calibration methods.

Our experiments show good agreement between the $\delta^{13}C$-$CH_4$ calibrations, but a systematic offset in the calibrations of $\delta^2H$-$CH_4$. While the average offset in $\delta^{13}C$-$CH_4$ of $+0.02 \pm 0.08$ ‰ is smaller than the measurement uncertainty, the average offset

in $\delta^2H$-$CH_4$ accounts for $+4.0 \pm 1.1$ ‰, which is significant.

The measurements of "Biogenic" for $\delta^{13}C$-$CH_4$ showed a larger discrepancy between the two methods. The cause of this discrepancy could not be resolved. It may be due to an unknown impurity in the bulk $CH_4$ gas that only impacts on the analysis of the pure $CH_4$ gas as $i$SAAC separates $CH_4$ chromatographically. A sudden drift in the "Biogenic" cylinder could be another reason for this discrepancy, however, this is not reflected in the analysis of

"Biogenic" as pure $CH_4$ gas over time. A pressure drop in the cylinder that is independent of $CH_4$ consumption during analysis is not detected. Because the discrepancy exceeds the measurement uncertainty by a factor of 4, we consider this result as an outlier and refrained from using it in further calculations.

Table 4: Comparison of the previous isotope scale for $CH_4$ in air of the $i$SAAC system as defined by "Carina-1", which was

calibrated by IMAU (Brass and Röckmann) and the new scale as based on the $CH_4$ calibrations presented in this study. The first column shows the name of the pure $CH_4$ gas that has been diluted with $CH_4$-free air for analysis as atmospheric sample (see Table 3). Columns 2 and 3 show the difference $\delta_{i\text{SAAC}} - \delta_{\text{pure}}$ for $\delta^2H$-$CH_4$ and $\delta^{13}C$-$CH_4$, respectively. The bottom line shows the mean difference with $1\sigma$ uncertainty. The cause for the larger value in $\delta^{13}C$-$CH_4$ of "Biogenic" is unknown. Therefore, this value is marked with (°) and is not considered for the determination of the scale difference.

| Gas name | $\delta^2H$-$CH_4$ [‰] | $\delta^{13}C$-$CH_4$ [‰] |
|---|---|---|
| Megan | 3.8 | 0.00 |
| Merlin | 4.0 | −0.01 |
| Minion | 2.7 | −0.03 |
| Melly | 4.3 | 0.13 |
| Mike-1 | 5.7 | −0.06 |
| Martha-1 | 3.2 | −0.05 |
| Fossil | 5.1 | 0.14 |
| Biogenic | 3.0 | 0.40 (°) |
| Average | +4.0 ± 1.1 | +0.02 ± 0.08 |




## 4 Discussion

We have used standard on-line IRMS techniques to calibrate pure $CH_4$ for $\delta^2H$ and $\delta^{13}C$ on international isotope reference scales. The pure methane gases were injected into the same isotope analysis system as water and carbonate reference materials and, thus, subjected to the same analytical conditions.

The oxidation of $CH_4$ to $CO_2$ (and $H_2O$) is usually accepted to produce no artefact. However, chemically $CH_4$ is rather stable; it requires high temperatures and a surplus of oxygen in order to drive the combustion reaction to completion. A major complication arises when yields are lower than 100 % (Merritt et al., 1995). This would allow for unreacted $CH_4$, which is a potent source of protonation in the IRMS ion source (Anicich, 1993). With $CO_2$ present, this results in the formation of $CO_2H^+$, an important isobaric interference on the $m/z$ 45 mass position, where $\delta^{13}C$ is measured. This source of analytical error is

important especially for analytical systems that don't separate $CO_2$ and $CH_4$ after the combustion on a chromatographic column or by cryogenic means. In the MPI-BGC systems, we use post-combustion chromatographic column to separate $N_2$ and $CO_2$ peaks, which also separates any residual methane from $CO_2$. In addition, we carefully checked for traces of methane left over from the reaction by monitoring the $m/z$ 15 ion current carefully for un-converted $CH_4$ and found the combustion to be quantitative.

The introduction of carbonates to quantitatively release the $CO_2$ at high temperatures has been demonstrated to yield high precision results (Coplen et al., 2006). A considerable advantage is that the oxygen isotopic composition is altered completely. By "roasting", it is forced to be very similar for all samples introduced into the combustion furnace, thus making the $^{17}O$ correction identical as well. Hence, any ambiguity arising from the necessary correction for extracting $\delta^{13}C$ values from the $m/z$ 45 ion current tends to cancel. The technique has been used to revise the VPDB scale by adding LSVEC as a second

scaling point (Coplen et al., 2006).

For hydrogen, we chose an analogue approach to process $H_2O$ and $CH_4$ through the high-temperature reactor of the TC/EA-IRMS system. This technique has been investigated earlier in detail and shown to result in quantitative conversion of $H_2O$ and methane at reactor temperatures above 1350 °C (Gehre et al., 2004). Possible artefacts can arise mainly from the different surface activities of $H_2O$ versus $CH_4$ before the conversion to $H_2$ (and CO or carbon) occurs. For water, this can lead to memory

effects, which needs to be taken into account quantitatively either by correcting for it or by discarding initial injections (Werner and Brand, 2001). We found a further, minor dependence of $\delta^2H$-$H_2O$ to the septum temperature. In the appendix we describe experimental details to ensure optimal conditions for making the $H_2O$ and $CH_4$ high temperature reactions directly compatible for quantitative $\delta^2H$ assessment and calibration.

We achieved the results presented in Table 3 for "Fossil" and "Biogenic" with a large number of analyses. Both gases had

already been calibrated by one of the authors in an earlier study using the more classical approach, i.e. combusting pure methane quantitatively off line and sampling the resulting $CO_2$ and $H_2O$ for consecutive isotope analysis (Sperlich 2012). In this study, the $CH_4$ derived $CO_2$ was analysed by dual inlet IRMS while the $H_2O$ was analysed by either TC/EA-IRMS or cavity-ring-down spectroscopy. The results of both calibration methods agree in the order of the combined uncertainty of both

methods for $\delta^2$H-CH$_4$ and $\delta^{13}$C-CH$_4$ (Table 3). The values from our new calibration for "Biogenic" and "Fossil" appear to be slightly more depleted in $\delta^2$H-CH$_4$ (average difference of 2.1 ‰) and in $\delta^{13}$C-CH$_4$ (average difference of 0.17 ‰), suggesting that the discrepancy could be systematic. We conclude from the broad agreement with the previously published results on "Fossil" and "Biogenic" that our new method is capable to calibrate CH$_4$ gases accurately to the international isotope scales.

Our new results are based on a large number of individual CH$_4$ conversions and measurements, using GC-IRMS methods that are used for routine analysis since more than a decade. In comparison, the previously published values of Sperlich et al., (2012) were derived from only four analyses per gas, converted in an off-line reactor. We suggest that the method presented here is more accurate and robust compared to the method of Sperlich et al., (2012).

The calibration of "Merlin" against "Megan" as Master-CH$_4$ as well as against primary reference standards provides another

indication of the accuracy of our analytical methods. The difference between the two calibrations of "Merlin" accounts for 1.6 ‰ in $\delta^2$H-CH$_4$ and for 0.07 ‰ in $\delta^{13}$C-CH$_4$, which is of the order of the analytical uncertainty. We interpret the good agreement as indicator that the presented methods for CH$_4$, H$_2$O and carbonate analysis do not create any significant isotope fractionation and suggest that our CH$_4$ gases are accurately calibrated. Note that this conclusion is a best estimation scenario that cannot be tested further without CH$_4$ reference material.

The total propagated uncertainty of the isotope calibration is smaller than or similar to the uncertainty of most analytical systems to measure CH$_4$ isotope ratios in air or ice core samples and can therefore help to increase the compatibility between international laboratories.

**5 Conclusions**

The number of laboratories that measure isotope ratios of atmospheric CH$_4$ is growing and combining data from multiple laboratories could lead to more powerful interpretative approaches of the combined data sets. However, the analysis of data from multiple laboratories is currently hampered by the lack of reference materials that enable the community to produce a unified data set.

To solve this problem, we accurately referenced twelve pure CH$_4$ gases to the international isotope scales for carbon and hydrogen and diluted aliquots of eight calibrated CH$_4$ gases with CH$_4$-free air in 5-L glass flasks. These synthetic gas mixtures were tested for their use as standards for CH$_4$ isotope ratios by comparing separately established isotope scales. From the combined results, the inter-laboratory calibration offsets of the two scales could be established reliably.

Our synthetic atmospheric reference gases for the isotopic composition of CH$_4$ will be made available to the atmospheric

monitoring community upon request. Using these reference gases in multiple laboratories will help unifying the data sets of the atmospheric monitoring community, thus enabling compatible isotope ratio data sets of atmospheric methane on a global scale over time.



### Acknowledgements

Financial support was provided within the European Commission projects InGOS and IMECC. N. A. M. Uitslag's visit in Jena was made available by the Erasmus program. Support by Huilin Chen and Harro Meijer (U-Groningen) is gratefully
appreciated. We are indebted to Ingeborg Levin for making archived air samples from Neumayer station available for comparison measurements and to Gordon Brailsford for valuable comments during the preparation of the manuscript.

### Appendix A: Preliminary experiments for calibration of $CH_4$ with $H_2O$ for $\delta^2H$

The injection of $H_2O$ samples into the reactor is critical because it is prone to isotope fractionation (Werner and Brand, 2001).
This isotope fractionation is caused by system memory due to adhesion of injected $H_2O$ to the reactor walls. The isotope fractionation can be overcome by repetitive injections of $H_2O$ samples with identical isotopic composition, thereby overwriting the memory effect until it reaches a marginal level. For $H_2O$ analyses under constant analytical conditions (e.g. constant reactor temperature), the adhesion effect is a function mainly of the amount of injected $H_2O$ sample. However, the effect on the isotopic composition furthermore scales with the isotopic difference between two consecutive samples (Gehre et al., 2004). Because
there is no adhesion of the sample during $CH_4$ analysis, this memory effect is specific only to the analysis of $H_2O$ in our study. Therefore, system memory of $H_2O$ could propagate into the $CH_4$ calibrations. Memory effects are identified in a series of replicate $H_2O$ measurements and are corrected for by modelling the memory function as described in Gehre et al., (2004) and Brand et al., (2009a) on a routine basis, as our system has been used for isotope analysis of $H_2O$ samples for more than a decade. We conclude that our results are free of artefacts from memory effects.
Isotope fractionation during the analysis of the reference waters can also be caused by insufficiently heated septa (Gehre et al., 2004). We injected 106 $H_2O$ samples while we increased the septum temperatures in 9 steps from 76°C to 137°C. In general, we observed a $\delta^2H$ enrichment with increasing Septum temperature, however, there seems to be a second effect in the course of increasing temperature. A systematic variation is apparent at temperatures above 90°C with stabilising $\delta^2H$-$H_2O$ values at septum temperatures around 130°C (Fig. A1, blue circles). At the three highest temperatures, $\delta^2H$-$H_2O$ averages at -62.6 ± 0.5
‰. The stabilising $\delta^2H$-$H_2O$ at high temperatures suggests quantitative $H_2O$ processing without significant isotope fractionation, in line with (Gehre et al., 2004). Note the disagreement with the three $\delta^2H$-$H_2O$ values at the lower temperature range (red diamonds) that show a matching slope but an offset that does not align with the $\delta^2H$-$H_2O$ pattern observed at temperatures above 90°C. We cannot explain this offset but can only speculate that the first experiments with a heated septum had a stronger initial impact on the measured isotopic composition. Note that the total septum temperature effect is in the order
of twice the typical measurement uncertainty and is thus difficult to observe.



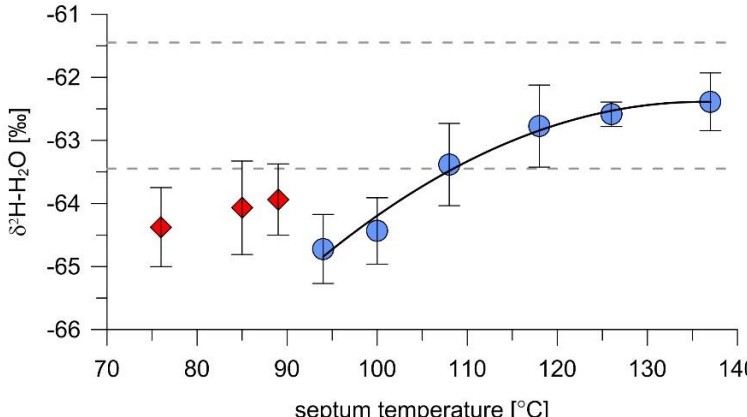

Figure A1: The $\delta^2H$ variation of $H_2O$ injections with septum temperatures. Blue circles show average values that fall onto a polynomial fit while red diamonds indicate the three values at low septum temperatures that include the offset. The error bars show 1σ standard deviations and the grey-dashed lines indicate the typical precision limit of 1 ‰ for $\delta^2H$-$H_2O$ analysis (Gehre et al., 2004) around the $\delta^2H$-$H_2O$ of -62.45 ‰ as estimated by the polynomial fit for the operating septum temperature of 130°C.

Quantitative conversion of both $CH_4$ and $H_2O$ in the high temperature reactor is of utmost importance for our study, because incomplete conversion causes isotopic fractionation in the reaction products (e.g. Burgoyne and Hayes (1998), Hilkert et al., (1999) Gehre et al., (2004)). The temperature of the reactor is critical for the efficiency of the conversion process. We performed an experiment with $CH_4$ and $H_2O$ injections at different reaction temperatures and show the results in Fig. A2. For water injections we observe a pronounced, non-linear $\delta^2H$-$H_2O$ change of ~15 ‰ with reactor temperature increase from 1300°C to 1450°C, reaching a plateau above 1400°C. This pattern is consistent with previous observations in both trend and magnitude (Gehre et al., 2004). In contrast, the linear fit for $\delta^2H$-$CH_4$ increases by only about 1 ‰ over the 150 K temperature range. However, the slope is statistically insignificant as shown by the 95 % confidence interval of the linear fit (Fig. A2). This analyte-specific isotope variation is also reflected in the area of the $H_2O$ and $CH_4$ derived $H_2$ peaks (Fig. A2). While the $H_2O$-derived $H_2$ peak areas increase with increasing reactor temperature, the $CH_4$-derived $H_2$ peak areas remain constant within the error bars throughout the experiments. For an unknown reason, three out of six $H_2$ peaks that resulted from $H_2O$ injections at 1400°C were by 10-15 standard deviations smaller than the remaining three peaks. We present the averages and 1σ standard of the $H_2$ peaks with and without removal of these outliers in Fig. A2, which shows the exceptional pattern at 1400°C. Despite of this peak size variability, the isotopic composition of all $H_2O$ injections at 1400°C is in good agreement. Our experiments indicate that high reactor temperatures of 1450°C are required especially for quantitative conversion of $H_2O$, while the effects of reactor temperature on both yield and the isotopic composition of $CH_4$-derived $H_2$ are comparably small. Therefore, we





operate the reactor at a temperature of 1450°C to guarantee quantitative conversion without isotope fractionation of both, $H_2O$ (Gehre et al., 2004) and $CH_4$ (Burgoyne and Hayes, (1998), Hilkert et al., (1999)).

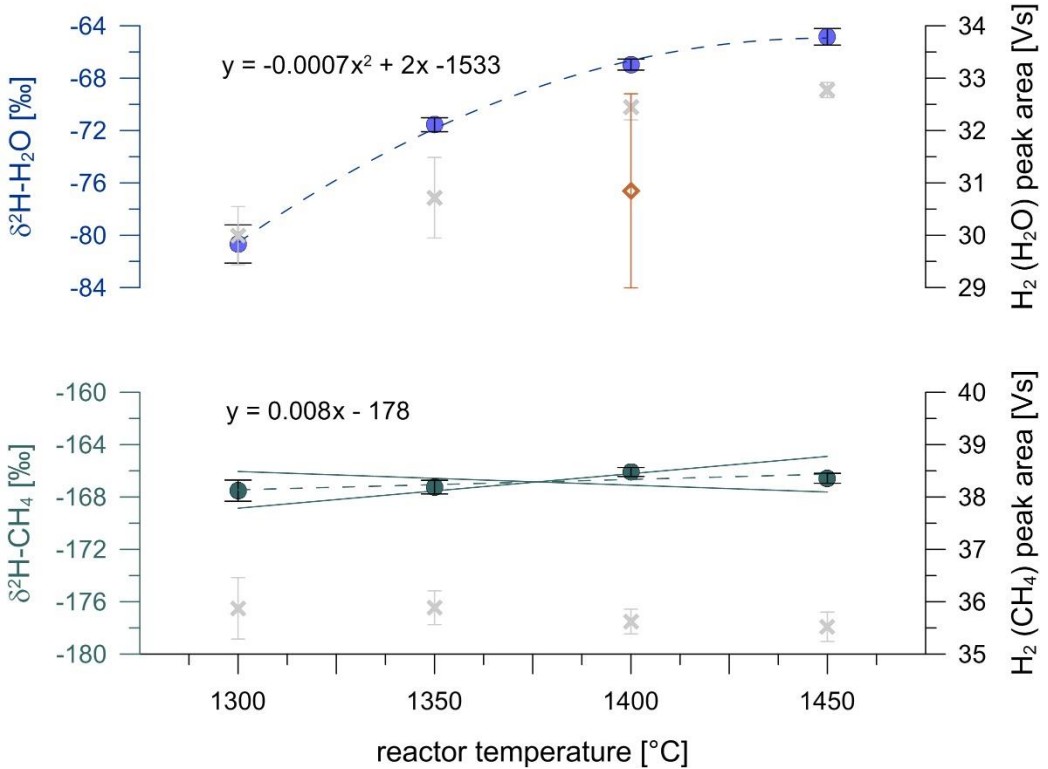

Figure A2: The variation in isotopic composition and peak area of $H_2$ derived from $H_2O$ and $CH_4$ injections at reactor temperatures between 1300-1450°C. Top and bottom panels show experiments for $H_2O$ and $CH_4$, respectively. Isotope ratios are shown in blue for $H_2O$ and green for $CH_4$ and refer to the left hand axes. Average $H_2$ peak areas are indicated by grey crosses and refer to the right hand axes. All error bars indicate the $1\sigma$ standard deviation. The red diamond shows the average peak area and respective $1\sigma$ standard deviation when the three outliers are included (see Appendix text). Y-axes ranges are

matched between top and bottom panels to enable direct comparison of the temperature effect for $H_2O$ and $CH_4$. Equations describe the fits in both panels, displayed by dashed lines. The continuous lines in the bottom panel indicate the 95 % confidence interval of the linear fit.

## Appendix B: Experiments for calibration of $CH_4$ with $CaCO_3$ and $Li_2CO_3$

Incomplete $CH_4$ combustion results in measurement artefacts, because $CH_4$-derived $CO_2$ and unconverted $CH_4$ enter the IRMS simultaneously, enabling the formation of $CO_2H^+$ fragments that contribute to the signal on the $m/z$ 45 detector (Werner and



Brand, 2001). In order to test the completeness of $CH_4$ combustion, we set the IRMS to monitor *m/z* 15 during the analysis of a $CH_4$ sample. We estimate that <0.1 % of an injected $CH_4$ sample reaches the IRMS unconverted, suggesting quantitative combustion of $CH_4$.

The analysis of carbonates can suffer from a well characterised blank contribution that is due to the carbon impurities within

the tin capsules that are used for the aliquotation of the carbonates (Werner et al., 1999). In contrast, no such blank is expected when samples are analysed without tin capsules, as would be the case for gaseous $CH_4$ samples. In order to fulfil the principle of identical treatment between analyses of carbonate reference materials and $CH_4$ samples, we added empty tin capsules to each $CH_4$ analysis and applied the routine blank correction to all measurements.

To test for the completeness of carbonate digestion, we added small amounts of tungsten trioxide ($WO_3$) to the carbonate

samples during weighing (about 1:1 by weight) to increase the instantaneous reaction temperature and to provide additional oxygen during the liberation of $CO_2$ from different carbonates. The addition of $WO_3$ did improve the peak shape for the analysis of $BaCO_3$ (Table 2) but had no effect on the results of our isotope ratio measurements. Other carbonates ($CaCO_3$ and $Li_2CO_3$) did not suffer from a broadened peak shape. This suggests that the carbonate digestion is not limited by either temperature or oxygen availability. Therefore, we conclude that the digestion of carbonate samples is quantitative and we refrained from

adding $WO_3$ during the $CH_4$ calibration measurements.

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
