# Peer review of "Development and evaluation of a suite of isotope reference gases for methane in air"

_Atmospheric Measurement Techniques, 2016_

## Referee Comment (RC1) · Anonymous Referee #1 · 29 Feb 2016

Comments to "Development and evaluation of a suite of isotope reference gases for methane in air" by Sperlich et al.

**Summary of manuscript**

The authors set up measurement systems for anchoring $\delta^{13}$C-CH$_4$ and $\delta^2$H-CH$_4$ to the international reference isotope scales. They also prepared a suite of isotope reference gases (pure CH$_4$ gases and synthetic CH$_4$-in-air gases) and calibrated them. The synthetic CH$_4$-in-air gases will be made available for intercomparison to achieve high compatibility in the atmospheric monitoring community.

**General comments**

This study serves a fundamental contribution to the research community in study of the global CH$_4$ cycle. Increasing number of data of CH$_4$ isotopic ratios in air is becoming available, especially since the development of continuous-flow measurement system. However, datasets from different laboratories cannot be currently merged due to unnegligible and unidentified inter-laboratory calibration offsets. This hampers optimized use of isotope data for better understanding of the CH$_4$ cycle. This study is a start up of anchoring measurements of different laboratories. The authors set up the current best available measurement methods, systems and calibration strategies. Further intercomparison efforts to distribute the synthetic CH$_4$-in-air standard gases developed in this study to worldwide laboratories are also an important next step. This study has significance with good measurement quality, and the manuscript is fairly well organized; however, many parts are not written in a clear and concise way and do not contain complete information. In my opinion, this study should be definitely published but the current manuscript version is not suitable for publication. I realize that this manuscript will be an important long-term reference for the research community. This is why I think that the manuscript should meet high level of transparency for all detailed information and clearness in description to help following researchers. It is regrettable that number of my comments below includes points that could have been thoroughly considered and corrected with the authors' responsibility before submission. I would like to encourage the authors to complete all details and rewrite the whole manuscript to improve the readability, after which this manuscript might be evaluated again.

It is very important to unify terms throughout the manuscript (not to mix up

different terms for same meaning). In particular, I suggest to clearly define hierarchy of the gases at early part of the manuscript and always use the terms defined. Name of the gases are bracketed by "" (e.g. "Megan"), but I do not think this necessary. I also suggest modifying Figure 1 so that readers easily find which gas is at which level (exact correspondence between Figure 1 and Tables 1 and 2). You might consider a figure similar to the attached figure and get it merged into Fig. 1 (see also specific comments to Fig. 1). This would improve traceability. To my understanding, the hierarchy is (1) IAEA "reference materials", (2) MPI-BGC "working standards" ($H_2O$ and carbonates), (3) "master" $CH_4$ gases (or "MPI-BGC primary" might be an alternative), (4) "secondary" $CH_4$ gases—these are also called "calibration" in tables, but I think "secondary" (or "MPI-BGC secondary") used at other places fits better, and (5) "synthetic $CH_4$-in-air standard" produced by dilution of the secondary.

Since iSAAC system plays an important role, the authors should elaborate at least overview of the system (e.g. materials and temperature of preconcentration traps, separation columns, etc.). Is it similar to Brass and Röckmann (2010)? There are many references citable. The authors might briefly describe the system even if they consider a separate publication. One more point on iSAAC—it seems that iSAAC is operated with "Carina-1" with nominal values on the IMAU scale, but this is not written explicitly. The authors should clearly describe this in the manuscript.

The IMAU scale is in the manuscript considered to be an established scale independent from the MPI-BGC scale developed in this study. Description of origin of the IMAU scale is also important for perfect peace out of the manuscript. I suppose that the IMAU scales (both $\delta^{13}C$ and $\delta^2H$) are also ultimately referenced to some of the IAEA reference materials. The authors should mention to this, not only cite to the technical paper Brass and Röckmann (2010). I understand that, as appeared in this study, during the course of the long history, systematic offsets could arise due to different measurement methods, different maintenance strategies etc. even that both scales are on the identical VPDB and VSMOW scales in theory—that is, the problem is in calibration at individual laboratories. It is important to describe this fact. This also helps future intercomparison planned.

Also confusing is that the gas "Carina-1" has $\delta^{13}C$ and $\delta^2H$ values on the IMAU scale. The authors might present those values with a note that they are on the IMAU scale, since they are listed in Table 2.

**Specific comments**

P2 L2: "Isotope ratios of trace gases" to "Isotope ratios of $CH_4$".

P2 L2: I do not think isotope ratios are still "an emerging tool"—they have tested over decades.

P2 L3: Quay et al. (1991, 1999) are also important studies that showed quantitative use of isotopic compositions for understanding the global $CH_4$ cycle, which could fit this context.

P2 L16: "papers" to "the paper"

P2 L17ff: The authors might list the IAEA reference materials used in the cited references. Also, the authors might mention to the fact that most (or all?) calibrations in the cited references were linked to the IAEA certified materials and that in theory on identical scales. It would highlight the fact that calibration offsets have arisen during scale propagations in individual laboratories—this probably corresponds to more ambiguous word "the diversity of referencing trajectories" (P2 L21) in the manuscript.

P2 L21: First, here in the context the authors should refer only to papers already published and thus "W.A Brand pers. comm." should be left out; second, personal communication with one of contributing authors is strange—same for other places. An alternative might be "W. A. Brand, unpublished data" if this matches the context. Otherwise unpublished results should not be cited unless clear necessity—it would not help readers and traceability of the manuscript.

P2 L22: "referencing" to "calibration"

P2 L23: I suggest to start this paragraph with a phrase like "For isotope ratios of $CO_2$ in air, …", so that readers better understand that the authors intend to introduce an analogical method already applied for an other gas.

P3 L7ff: Please unify use of "the Principle of Identical Treatment" and "PIT" everywhere in the manuscript.

Figure 1: As pointed out in the general comments, the authors might enrich this figure by adding information on which standards in Table 1 are at which level in this figure. An other idea is to split this figure both for $\delta^2H$ and $\delta^{13}C$ as the figure below. Another good reference is Figure 2 of Sperlich et al. (2012).

[Figure]

P3 L15: I do not think that "commercial provider", "methanogenic origin" and "isotopic origins" should be listed in parallel with equal stress. Differences in methanogenic origin and isotopic composition in $CH_4$ is highly linked; the former is a primary factor for the latter. Roughly speaking, commercial providers do not always care about it.

P2 L17: "and are therefore of known isotopic composition" should be left out. The "Biogenic" and "Fossil" gases are treated as unknown in this study and removing the phrase would reduce risk of misunderstanding.

Table 1: I am not sure if "create" is a suitable word. I think "produced" sounds more common.

P4 L7: The contents in the first paragraph of section 3.1 should be moved here.

P4 L15: "We show the most recent values…" The table gives references, but the caption mentions only to Brand et al. (2014)—this is confusing. If original papers are cited in the table, I would delete this sentence of the caption.

P5 L4: The abbreviation "IRMS" appears here for the first time.

P5 L6: "after the conversion to $H_2$"—since there is "to covert $CH_4$ to $H_2$" just a one line above, this phrase is a redundancy. This sentence may be like "…to convert $CH_4$ to $H_2$ (+carbon) for subsequent measurement of $\delta^2$H-$CH_4$ in pure $CH_4$ gases on IRMS."

P5 L4–L9: In my first reading, I (probably) misunderstood that $CH_4$ and $H_2O$ in sample are converted to $H_2$ in different reactors. Besides the above suggestions, I suggest possible reorganization of the sentences: "We use a TC/EA coupled to an IRMS via an

open split for $\delta^2$H measurements of CH$_4$ and water; CH$_4$ in sample gas is introduced into the TC/EA and converted to H$_2$ in a glassy carbon reactor maintained at 1450°C; water sample is injected through a heated septum at 130°C into the same reactor and converted to CO and H$_2$; the converted H$_2$ is measured on the IRMS for $\delta^2$H."

P5 L9: "hydrogen isotopic composition" to "$\delta^2$H"

P5 L11: "that is configured as shown in Fig. 2" to "configured as Fig. 2"

P5 L12: "Typical CH$_4$ feed flow rates range between 2-3 mL/minute." to "Typical CH$_4$ feed flow rate is 2–3 mL/minute." How much volume of CH$_4$ is actually injected?

P5 L14: Leave out "now". My suggestion is like: "Measurement sequences are configured by combining injections of CH$_4$ gas and reference water materials into the reactor, which are via the 10-port valve (Fig. 2) and septum from an autosampler, respectively." I would leave out ", where both CH$_4$ and H2O are converted…"

P5 L16: I would leave out "Apart from the injection procedure…" because it comes up later soon again.

P5 L18–19: "The amounts" to "Amounts". "to achieve matching peak…" to "to match peak…". "during IRMS analysis" to "of the IRMS output signal". But this sentence might be left out because same (and more in-detail) explanation appears page 7.

P5 L20: "Fig. 4" to "Fig. 3"

P6 L8: "This configuration enables alternative injections of CH$_4$ gases with known and unknown $\delta^2$H values in an identical fashion; the calibrated "Megan" and "Merlin" served as known reference gases for calibrations of the other CH$_4$ gases (Table 1)." I do not get what "the respective isotope scales" means.

P6 L14: "applying" to "employing"

P6 L15: "working reference waters" to "working standard" because they seem to be abbreviated as "ws" in Table 2. Decapitalize www-"J"1 and BGP-"J"1 to harmonize with Table 2.

P6 L20ff: use "PIT" throughout the manuscript if this term is abbreviated.

P7 L4: leave out "several", otherwise write more explicitly.

P7 L8: "assumed to be" kept constant

P7 L13: "…, which guarantees same level of H$_3$-factor correction and allows use of the standard integration software (ISODAT, Thermo, *Company name & place as other places*)"

P7 L14–18: "We performed…" Please write explicitly about what were tested. I do not

understand which experiments in appendix correspond to these descriptions. I could not find descriptions that support these sentences—it sounds like that the authors argue good performance of their measurements without showing any clear evidences.

P9 L5: "routinely used"—does this mean the system routinely used at MPI-BGC or at other laboratories? This sentence may intend to endorse reliability of measurements in this study, but it would not work if the authors mean the latter. Even if the former, just "routinely used" cannot justify. I would write more explicitly—for instance, "Similar systems were used for… in previous studies (…et al.)" or "MPI-BGC has operated this system for… over X years (…et al.)." The same comment is also for P5 L6.

P9 L7: My suggestion: "Outflow of the 10-port valve enters into a combustion furnace with helium carrier gas stream of 10 mL/min through an 1/16 inch tubing (70/30% Cu/Ni alloy) specially fitted to the EA system (Fig. 4)." How important is "the oxygen plume region"? If this is critical, elaborate more. Otherwise the following sentence may be enough.

Figure 4: "oxidation"—the text use different terms like "combustion chamber" and "combustion reactor". This confuses readers. "combustion furnace" (also "reduction furnace" might be good.

P9 L13: "the combustion reactor"—is this different from "the combustion chamber" in the preceding sentence? Use one term if not.

P9 L14: "All samples are oxidized to $CO_2$ in the combustion furnace maintained at 1020° C (Werner et al., 1999) and experience identical analytical treatment (PIT) thereafter." I think that the authors monitor furnace's temperature but not temperature at which sample actually reacts. Do not mix up "combustion" and "oxidation" so often, otherwise readers wonder if the authors intend to use them with different meanings.

P9 L16: "The sample is dried by passing through a Nafion dryer (…) and a trap filled with $Mg(ClO_4)_2$, and then introduced into a GC column (3 m×1/4 inch Porapak PQS, CE instruments) held at 80° C."

P10 L1–3: Circumlocution. My suggestion: "By alternating injection of $CH_4$ gas and carbonate reference materials such as LSVEC, Mar-j1 and ali-j3, we referenced our Master $CH_4$ gas (Megan and Melrin) to the VPDB isotope scale." Later NBS-19 also appears (not listed in Table 2), and the authors describe Megan and Merlin were calibrated against NBS 19 and LSVEC. How were Mar-j1 and ali-j3 used? Were they used to assign values to the Master gases? Or just for measurement control? Clarify this.

For $\delta^2H$ calibrations, the authors used "in house standards" to assign values to the Master gases, but here for $\delta^{13}C$, the Master gases are directly measured against the IAEA materials. I do not understand what "over three independent periods each" means.

P10 L5: "analysis" to "analyses"

P10 L13–P11 L1: "The first three square-shaped peaks and the last tailing peaks are for pure $CO_2$ working gas and $CH_4$- and $Li_2CO_3$-derived $CO_2$, respectively."

P11 L2: "…(with peak widths of 101 s and…)

P11 L7: "the two-step calibration strategy approach"—Clarify meaning of this. Does it mean the calibration has two anchoring points on the VPDB (VSMOW) scale?

P11 L8: As mentioned earlier, NBS 19 appears here for the first time and not listed in Table 2. Megan and Merlin are bracketed by "" at other places, but not here.

P11 L14: "…has been found, which has been…"

P11 L18: "BGC ISOLAB"—Would not "MPI-BGC" work as other places?

P11 L27: "The secondary or "calibration" $CH_4$ gases"—as mentioned earlier, define the hierarchy of standards early in the manuscript so that confusing words like here do not come up.

P11 L30: Which is correct "Master $CH_4$ gas" or "master methane gas"?

P12 L1ff: I suggest to use a term "synthetic $CH_4$-in-air standard" for gases produced by dilution of the secondary $CH_4$ gases.

P12 L2: Same comments as P11 L18.

P12 L3: "This system (named ARAMIS) is used to produce synthetic standard gases with atmospheric $CH_4$ mole fraction levels."

P12 L4: "We diluted aliquot of the secondary $CH_4$ gases with $CH_4$-free air to…"

P12 L5–6: "The $CH_4$-free matrix air, which was produced by target-mixing ultra-pure constituents, contains $N_2$, $O_2$, $N_2O$ and Kr at atmospheric levels, so that composition of the produced gas is as close to ambient air as possible."

P12 L7–8: "…to account for the interference effect on the $\delta^{13}C$-$CH_4$ measurements using GC-IRMS systems (Schmitt et al., 2013)." I would leave out "so that…" but add this part to the preceding sentence.

P12 L8: "A sensitive analysis…" I do not understand what this sentence means. An upper limit for what? Sensitive to what? What is high-precision gas-chromatography? Is it different from GC-IRMS?

P12 L9: I do not think Table 1 gives "further details."

P12 L11: "an average standard deviation" to "average standard deviations", but what does an average standard deviation mean? Standard deviations for measurements of synthetic standard from multiple dilutions were averaged?

P12 L12: Personal communication with one of contributing authors is odd. For this sentence, the authors should give complete description of measurement precisions of the analytical system in section 2.6.

P12 L17: Same comment as P12 L12.

P12 L16–L20: As described in general comments, the authors should describe at least overview of the iSAAC system. Since it seems to be a continuous-flow system, likely similar to Brass and Röckmann (2010), the authors should give configuration of the system with basic information of key components, so that at least relevant researchers can understand the system well, because this system is a key to calibrate the synthetic $CH_4$-in-air standards.

P12 L22: As described in the general comments, even that readers are led to Brass and Röckmann (2010) for very details, the authors should describe basics of the IMAU scale. That is, with what types of laboratory standards they maintain long-term consistency of their scales, how and to what IAEA materials their scales are ultimately referenced. As I mentioned earlier, their scales are also the VSMOW and VPDB scales; the IMAU scales are identical to the MPI-BGC scales in theory.

P12 L23: ""Carina-1" as master reference gas for the iSAAC system" means, as long as any gases are measured by iSAAC system, they are assigned values on the IMAU scale? This should be clarified and it would help understand Table 4 and relevant texts.

P12 L24–25: Are Carina-1 and Carina-2 of identical origin and should they agree both $\delta^{13}C$ and $\delta^2H$ in theory? This is unclear. Only the description "Jena air" in Table 1 does not guarantee it. Regarding the offset in $\delta^2H$, was the cause identified? Also, the authors might present $\delta^{13}C$ and $\delta^2H$ values of Carina gases on the IMAU scales.

P12 L26: "The synthetic isotope reference gases" to "The synthetic $CH_4$-in-air gases", "previously calibrated $CH_4$ gases" to "the secondary $CH_4$ gases"

P12 L27: "The results of these measurements are compared to the calibration results of the secondary $CH_4$ gases so that the differences between the calibrations in this study and the iSAAC measurements against Carina-1 indicate the offsets between the MPI-BGC and IMAU scales."

P13 L4: "carbon and hydrogen isotope ratios" to "$\delta^{13}C$-$CH_4$ and $\delta^2H$-$CH_4$"

P13 L7: "in the reference hierarchy"; "calibration $CH_4$ gas" to "secondary $CH_4$ gas"

P13 L8: "Therefore" to "As a result"

P13 L8: "primary reference materials" to "international reference materials"; or the authors might use "RM" for the IAEA certified reference materials, with this stated in early part of the manuscript

P13 L9: "against a Master-$CH_4$ gas Megan"

P13 L11: "Both Megan and Merlin are fossil in origin with typical $\delta^{13}C$-$CH_4$ and $\delta^2H$-$CH_4$ signatures (e.g. …). The two Master-$CH_4$ gases are similar in $\delta^2H$-$CH_4$ with calibrated values of −168.0±0.6‰ for Megan and −165.7±0.6‰ for Merlin. The calibrated $\delta^{13}C$-$CH_4$ values are −40.75±0.07‰ for Megan and −39.07±0.07‰ for Merlin." Megan's $\delta^2H$ is −168.1 in Table 3, but −168.0 here in the text.

P14 L5: I would leave out lines "Fossil comparison" and "Biogenic comparison" from the table and bring them to footnote of the table. This might improve ease of viewing of the table by focusing only on calibration result of this study.

P14 L7: "were then mixed with $\delta^2H$-spike gas to produce Martha-2 and Mike-2 with $\delta^2H$-$CH_4$ values higher than or similar to that of the tropospheric $CH_4$, respectively."

P14 L11: "Results for secondary $CH_4$ gas calibrations against Master $CH_4$ gases"

P14 L13: "…spiked $CH_4$ mixtures, and thus cover wide range of $\delta^{13}C$-$CH_4$ (…) and $\delta D$-$CH_4$ (…)." I would leave out ", which include…"

P14 L15: "create" to "produce"

P14 L15: "a $CH_4$ gas with $\delta^2H$-$CH_4$ close to that of the tropospheric value, …"

P14 L16: "a fossil $CH_4$ gas"—Is this the "Fossil" gas? I wonder if "diluted" is the correct word, because dilution usually means lowering mixing ratio of certain compounds, but here the $CH_4$ gases are almost pure gases and mixture of such gases just result in no change in $CH_4$ mixing ratio.

P15 L1: "Results for calibrations of synthetic $CH_4$-in-air standards"

P15 L2: "Aliquots of the secondary $CH_4$ gases were diluted with $CH_4$-free air to produce the synthetic $CH_4$-in-air standards (section 2.5) for analysis on the iSAAC system (section 2.6)." Large part of L3 is redundancy.

P15 L4: "the diluted $CH_4$ reference gases" to "the synthetic $CH_4$-in-air gases"

P15 L4: "…against Carina-1 on the IMAU scale."

P15 L5–7: My suggestion: "We calculate the difference between the calibrations of the secondary $CH_4$ gases on our measurement systems (sections 2.2 and 2.3) and the

synthetic $CH_4$-in-air standards on iSAAC, $\delta_{iSAAC} - \delta_{sec}$ (Table 4); the value indicates calibration offsets between the MPI-BGC and IMAU scales, if we assume no isotopic fractionation in the dilution process."

P15 L8–9: My suggestion: "Our experiments show a good agreement for $\delta^{13}$C-$CH_4$ with an average difference of +0.02±0.08‰, but a significant systematic offset of +4.0±1.1‰ for $\delta^2$H-$CH_4$." What is the cause of the $\delta^2$H-$CH_4$ offset? The authors should discuss on possible sources of the offset here or in the next section.

P15 L13: "the pure $CH_4$ gas" to "the secondary $CH_4$ gas"

P15 L13: "A sudden drift" by what kind of reason?

P15 L19: "Comparison of the calibrations on the new MPI-BGC scale developed in this study and iSAAC measurements on the IMAU scale (Brass and Röckmann, 2010)."

P15 L21: "the name of the Master/secondary $CH_4$ gas that was diluted to the synthetic $CH_4$-in-air standard for iSAAC measurements"

P15 L23: "the mean difference" to "the average difference"—use one term both for caption and table.

P15 L24: "Therefore, this value (marked with °) was excluded for calculation of the average scale difference."

P16 L2: "…to calibrate the pure $CH_4$ gases for…"

P16 L3: "methane" to "$CH_4$"; "…into the isotope measurement system that also analyses water and carbonate reference materials, and thus subject to PIT."

P16 L5: "The online oxidation of $CH_4$ to $CO_2$ (and $H_2O$) is considered to produce no isotopic fractionation." This sentence needs references.

P16 L6: "However, $CH_4$ is relatively stable chemically; complete oxidation of $CH_4$ thus requires high temperature and surplus of oxygen."

P16 L7: "allow for" to "leave"

P16 L8: "$CO_2$ present" to "presence of $CO_2$"

P16 L9: "This source of analytical error" to "This effect"

P16 L10: "don't" to "do not"

P16 L11: "In the MPI-BGC systems" to "In the MPI-BGC EA-IRMS system"; Is this also the case for iSAAC? Clarify.

P16 L12: "methane" to "$CH_4$" (2 places)

P16 L14: "quantitative" Does it mean combustion efficiency of 100% or close to 100% without measureable isotopic fractionation? Write explicitly. Meaning of "quantitative"

is unclear also for many other places.

P16 L15: "quantitatively"—same comment as the above. I would write: "It has been demonstrated that introduction of carbonates into high-temperature combustion furnace yields CO$_2$ conversion resulting in high-precision $\delta^{13}$C measurements (…)."

P16 L16: "…oxygen isotope composition is altered completely in the conversion process from the original carbonate to the product CO$_2$."

P16 L17: I do not understand what this sentence means.

P16 L18: "ambiguity" to "uncertainty"; "extracting" to "calculating"; "values" to "value"; "tends to cancel" means that it is not 100% guaranteed and there are exceptions. The authors should better justify. Besides, this paragraph seems to be readable for only expert readers who can easily refer to equations for $^{17}$O correction. The authors should present "kind" introduction at the beginning of this paragraph on why this matters—I think this needed for AMT which expect readers more general than e.g. RCM.

P16 L21: "hydrogen" to "$\delta^2$H measurement"

P16 L22: "quantitative conversion"—same comment as the above.

P16 L23: "methane" to "CH$_4$"; My suggestion: "Major artifact can arise from more variable surface adhesion of H$_2$O than CH$_4$ in the combustion furnace before they are converted to H$_2$ (and CO/carbon)."

P16 L24: "water" to "H$_2$O"; I would write: "This can lead to memory effect in the $\delta^2$H-H$_2$O measurements, then corrections or discarding initial injections are needed (…)"

P16 L26: "In addition, we found a minor dependence of…"; "In the appendix" to "In Appendix A"

P16 L29: "with a large number of analyses"—number of analyses is given neither in Table 3 nor text. Without this, this description is not justified. I would write instead: "We have presented calibration results of the secondary CH$_4$ gases Fossil and Biogenic (Table 3). Both gases…"

P16 L30: Leave out "in an earlier study"; "the" to "a"; "combusting" to "combustion of"

P16 L31: "methane" to "CH$_4$"; "sampling of"; "consecutive" to "subsequent"?

P16 L31–33: "Sperlich et al. (2012) analyzed the CH$_4$ derived CO$_2$ for $\delta^{13}$C-CH$_4$ on a dual inlet IRMS and the CH$_4$ derived H$_2$O for $\delta^2$H-CH$_4$ on a TC/EA-IRMS system similar to this study or cavity-ring-down spectroscopy."

P16 L33: My suggestion: "Our calibration results are in overall agreement in both $\delta^{13}$C-CH$_4$ and $\delta^2$H-CH$_4$ with the previous values by Sperlich et al. (2012) within the uncertainties of both measurements (Table 3)."

P17 L1–3: "However, our calibration results for Biogenic and Fossil appear to…, suggesting systematic offsets."

P17 L3: This statement is strange. After this, the authors argue that calibrations by Sperlich et al. (2012) are less robust (which I do not think justified), but here the authors argue that their measurements are supported by agreement with the unreliable measurements by Sperlich et al. (2012).

P17 L5: "a large number of measurements"—same comments as P16 L29. I do not think that long-term use itself guarantees accuracy and robustness of a measurement system. Long-term use with unidentified artifacts can happen. The Kr interference on $\delta^{13}$C-CH$_4$ measurements is a good example—GC-IRMS had used for more than a decade until it was found.

P17 L6–8: I do not think that the statement in these sentences is justified. Sperlich et al. (2012) indeed has limited number of measurements, but did thorough treatments for complete combustion and reduction. Therefore, combustion and reduction by Sperlich et al. (2012) might be more complete than the online conversions made in this study. If so, the calibrations by Sperlich et al. (2012) might be more robust even if number of measurements is less. To keep the authors' argument, the authors should describe weak points of Sperlich et al. (2012) specifically.

P17 L11: "uncertainty" to "uncertainties"

P17 L12: "an indicator"; "create" to "cause"

P17 L14: I do not understand what the authors argue here. With "CH$_4$ reference materials" (which I do not understand what the authors refer to), what would the author do for further tests? What is the authors' best idea?

P17 L15–17: "The total propagated uncertainties in our calibrations are smaller than or similar to uncertainties of widely used analytical systems for $\delta^{13}$C-CH$_4$ and $\delta^2$H-CH$_4$ in air/ice core samples (references are needed). Therefore, a suite of standard gases developed in this study can help to increase the compatibility between international laboratories."

P17 L21: "The number" to "Number"

P17 L22: "could lead to" to "provides"; delete "of the combined data sets"

P17 L22–23: My suggestion: "However, such merged dataset has not been achieved by the lack of reference materials that enable direct intercomparison in the community."

P17 L25–26: The paragraph can just follow the previous one without line break; My suggestion: "To deal with this problem, we prepared 12 pure $CH_4$ gases (the secondary $CH_4$ gases) and accurately referenced them to the international isotope scales VSMOW and VPDB for $\delta^2H$-$CH_4$ and $\delta^{13}C$-$CH_4$, respectively. These secondary $CH_4$ gases then were diluted to produce 8 synthetic $CH_4$-in-air standards in 5-L glass flasks."

P17 L26–28: I do not understand what the authors argue here. The authors say the synthetic $CH_4$-in-air standards were "tested for their use", but where in the manuscript did they evaluated usability of the gases? Section 3.3 does not seem to describe this. "separately" to "indendently".

P17 L29: "synthetic atmospheric reference gases" to "synthetic $CH_4$-in-air standards"; "isotopic composition of $CH_4$" to "$\delta^{13}C$-$CH_4$ and $\delta^2H$-$CH_4$"

P17 L30: My suggestion: "These synthetic $CH_4$-in-air standards will help worldwide laboratories to anchor their measurement datasets to unified $\delta^{13}C$-$CH_4$ and $\delta^2H$-$CH_4$ scales shared in the atmospheric monitoring community, enabling compatible isotope ratio datasets for better understanding of the global $CH_4$ cycle."

P18 L8: This appendix describes "preliminary experiments", but the authors state in the main text that they optimized the measurement condition.

P18 L11: "This effect is avoidable by repetitive…"

P18 L13: "However" to "Moreover"

P18 L14: delete "furthermore"; "scales with isotopic difference" to "depends on difference of isotope ratio"

P18 L15: delete "only"

P18 L16: I do not understand how this sentence is liked to the preceding sentence by "Therefore".

P18 L17: delete "for" after "corrected"

P18 L18–19: "…, as our system"—as I mentioned for P17 L5, long-term operation itself does not guarantee correctness, so the last sentence of this paragraph is not justified.

P18 L21: "We made 106 injections of an identical $H_2O$ samples…"

P18 L22: "Septum" to "septum"; I would delete "however, there seems to…"

P18 L23: My suggestion: "A systematic increase of $\delta^2H$-$H_2O$ with the septum

temperature is apparent above 90° C, but the $\delta^2$H-H$_2$O value reaches the plateau around 130° C."

P18 L24: "At three highest temperatures"—The authors say the $\delta^2$H-H$_2$O value stabilized above 130°C, then the average should be calculated from the data above 130°C.

P18 L25: "The $\delta^2$H-H$_2$O values stabilized above 130° C suggests  adequate conversion of H$_2$O  without …"

P18 L26: "at the lower temperature range" to "below 90°C"; but my suggestion is for instance: "In contrast, the $\delta^2$H-H$_2$O values below 90°C show an insignificant slight increase with the septum temperature, which deviates from the pattern above 90°C." I do not understand the original sentence. What does "an offset" mean?

P18 L28–30: I do not understand these sentences. Elaborate better.

P19 L3: "fall onto a polynomial fit" is the authors' interpretation. Here the authors should write, for instance, as "the black line is the quadratic polynomial fit to the data above 90°C".

P19 L4: What is "the offset"?

P19 L6: The reasoning of taking a value from the polynomial fit is unclear.

P19 L10: "high temperature" to "high-temperature"; "utmost" to "particular"

P19 L12: I would leave out "The temperature…"

P19 L13: "…at different reactor temperatures (Fig. A2)."

P19 L16: "150 K" to "150°C"

P20 L14ff: I do not find where this appendix fits in the main text and what it supports.

---

## Referee Comment (RC2) · I. Levin (Referee) · 23 Mar 2016

**General Comments:**

First, I want to congratulate Sperlich and co-workers for their extremely valuable and important work, which will hopefully, more than three decades after the first publications of isotope measurements in atmospheric methane, solve our problem of lack of reference standards for these analyses. Sperlich et al. present a sound way for linking carbon and hydrogen isotope ratios in pure CH4 to the internationally accepted IAEA carbonate and water reference materials. After dilution of these calibrated pure CH4 gases with CH4-free synthetic air they produce CH4-in-air mixtures of ambient concentrations that can be used in the future as calibration standards, linking atmospheric (and source) methane isotope analyses from globally distributed labs to a common calibration scale.

As was already pointed out by Referee # 1, this fundamental work will become one of our basic references to describe the development of our future methane isotope calibration scale. As such, however, the descriptions of procedures in the current version of the manuscript, unfortunately, do not fully meet the requirements for clarity and completeness. Referee # 1 has already prepared a long list of comments and made very good suggestions for improvements of the manuscript, which I fully support. In my list of comments below, I thus only want to re-emphasize a number of points, which I feel most important to be tackled in a revised manuscript.

**Specific Comments:**

1. Introducing the various standard materials, their production (e.g. also by spiking with deuterated CH4), hierarchy and their calibration against IAEA reference materials (rm), or against other CH4 gases or other CH4-in-air gas mixtures is rather confusing. This does not only concern Figure 1 and Tables 1 and 2, but also Table 3, where the calibration results are given. I like very much the revised Figure 1 suggested by Referee #1. Please also be VERY clear with your nomenclature, e.g. distinguishing between "calibrations" (i.e. against reference materials) and what, from my point of view should better be named "comparison" with the earlier MPI-Jena standard gases Carina. In fact, it is not really clear to me how the H2 scale in Brass and Röckmann (2010), which forms the basis of the earlier MPI-Jena scale has been established. In their paper Brass and Röckmann refer to a paper by Bergamaschi et al. (1994) who obtained their calibration from colleagues at BGR, Hannover. The observed $\delta 2H$ difference of 4‰ between the IMAU/earlier MPI-BGC scale and the recent calibration may perhaps not be surprising. What does the remark on page 12 line 24-25 mean in this context? More information about the origin of the IMAU/earlier MPI-BGC scale is required to judge on the comparison results listed in Table 4.
   I think, the sentence in the conclusion P17, L 27 is too strong as the earlier MPI-BGC scale is only propagated from some yet unexplained origin.
2. Concerning the experimental set up for the calibrations against carbonates and water, I find the descriptions confusing and much too brief. A figure that displays the complete setups (for CH4 against carbonates and for CH4 against water) would be very helpful. Figures 2 and 4 could then be integrated there.
3. Discussion on accuracy of the calibrations: Although the authors have explained in detail how they tried to follow, as much as possible, the principle of identical treatment (PIT) and to avoid possible pitfalls when calibrating CH4 against carbonate and H2O reference materials, they cannot be sure that indeed no systematic biases have occurred. The most convincing argument for accuracy of the new standards to me is the good agreement with the earlier work by Sperlich et al. (2012) who used a (slightly) different procedure than in the present work. The discussion of the uncertainty in this respect is not clear enough. It seems rather to come as a mixture of long-term precision, agreement with the IMAU scale (see my reservations above) and finally arguing with "the combined uncertainty". I would like to see

here a more elaborated discussion and clear separation of the different indicators for accuracy. This could hopefully help to pin down biases in the future.

4. I also agree with Referee #1 that a description (or at least a reference to a publication) of the iSAAC measurement system is required.

**Minor comments:**

The Appendix is named Appendix 1 in the text but A and B when they show up

Sect. 3.2: that the high d2H values have been produced by spiking should go into section 2.1

In the discussion section it may be helpful to explain why m/z = 15 is used to detect unconverted CH4 in the sample.

---

## Short Comment (SC1) · 31 Mar 2016

Stable isotope measurements of greenhouse gases $CO_2$ and $CH_4$ make a powerful tool used to understand processes involved in the global carbon cycle. In order to get meaningful interpretation of stable isotope data in greenhouse gases, the data produced in different labs and in different years should be compatible within certain limits (WMO, 2014), these are called as compatibility goals (Table 1). In the last years compatibility of air-$CO_2$ stable isotope data is thought to be improved by introducing calibration "JRAS air" mixtures (WMO, 2012).

The compatibility goals for $CH_4$ are still a challenge to be achieved; it can be realized by using optimised calibration schemes, and to be based on appropriate reference materials with low uncertainty. All in all the compatibility goals can be considered in the first instance as the uncertainty required for "fit-for purpose" reference materials. In turn, such reference materials have to be compatible with a sample, that is why the community needs several reference $CH_4$-in-air mixtures.

**Table 1.** Compatibility goals for atmospheric CH4 (after WMO, 2014).

| Component | Compatibility goal (for background air) | Extended compatibility goal (for polluted air) |
|---|---|---|
| $\delta^{13}C$-$CH_4$ | ± 0.02‰ | ± 0.2‰ |
| $\delta^2H$-$CH_4$ | ± 1‰ | ± 5‰ |

Thereafter, work on the calibration of pure $CH_4$ gases aimed to produce reference $CH_4$-mixures cannot be published without thoughtful considerations of the uncertainty estimation and without clear presentation of the uncertainty budget. In particular this is expected for the work presented by the WMO-GAW Central Calibration Lab for stable isotopes in greenhouse gases (currently MPI-BGC, Jena, DE). In this respect the manuscript demonstrates serious problems such as unclear presentation of the calibrating approach in general, unclear uncertainty budget, as well as potentially missing/neglecting some uncertainty components. The major shortcuts are as following:

1. In order to build a skeleton of the uncertainty propagation, one has to consider a traceability chain for all measurements. The traceability chain has to be tracked to the highest Ref.Materials (RMs) in use. In case of $\delta^{13}C$ these are NBS19 & LSVEC (these have to be considered with their uncertainties) and include all measurement steps. Each next measurement step (including measurements on RMs) introduces an analytical uncertainty, thus increasing the total uncertainty.

2. The uncertainty propagation should be based on the traceability chain and also include all potential effects due to TC/EA, gas dilution etc. Besides, I would suggest to present the uncertainty budget, namely to describe a contribution of each uncertainty component starting from the uncertainty assigned to RMs carbonates (NBS19 and LSVEC), then the uncertainty of carbonate measurements, the uncertainty of 2-poit calibration as based on the carbonates, then analytical uncertainty of "master"-$CH_4$ (this is used for calibration "calibration"$CH_4$) etc. Such uncertainty budget will clearly demonstrate where further improvements are essential.

3. The "master" $CH_4$ (and its replacement when the first "master" was lost) was calibrated vs the IAEA Ref.Materials by applying the 2-point calibration approach. Next, several "calibration" $CH_4$ were calibrated vs the "master" $CH_4$. It is unclear how the 2-point calibration was applied in the case of measuring several "calibration" $CH_4$ gases? In fact

calibration vs. the "master" CH$_4$ looks like 1-point, thus violating the 2-point calibration approach (Coplen et al., 2006) designed to address various effects during sample preparation and measurements. I stress – this is in particular critical for $\delta^{13}$C values being down to -69.9 ‰ (Tab 3 in the manuscript), far below -40 ‰ of the "master"CH$_4$ and also outside the LSVEC value of -46.6 ‰.

4. Given that "calibration" CH$_4$ gases were characterised against the "master" CH$_4$, it is unclear why the $\delta^{13}$C-uncertaitnty of 0.06 ‰ for Martha-1 ("calibration"-CH$_4$) is smaller than the uncertainty of 0.07 ‰ obtained for the "master" CH$_4$. The uncertainty of each next material cannot be smaller than the uncertainty of material(s) used for its calibration (in this case uncertainty of "master" CH$_4$). This example implies something to be wrong in the uncertainty evaluation scheme in general. For the same reasons the $\delta^{13}$C uncertainty of ± 0.08 ‰ given for the "calibration" CH$_4$ Mike-1 looks like optimistically too low.

5. The authors should also explain the uncertainty values for "Biogenic" and "Fossil" CH$_4$, namely the values of ± 0.04‰, as given with the reference to (Sperlich et al., 2012). Why these are lower than uncertainties obtained by the work presented in this manuscript? In fact Sperlich et al. (2012) gave no detailed explanation on the uncertainty propagation. Given that the paper by Sperlich et al. (2012) is written by the same authors as the present manuscript, this is a must-requirement.

6. When focusing high accuracy values, the authors need to consider the effect $^{17}$O correction for the entire $\delta^{13}$C-calibration scheme, namely when calibration started from carbonates is applied to CH4 gases. Is there any potential bias?

7. Last but not least, the authors wrongly cite the $\delta^{13}$C-uncertainty of LSVEC. The message sent in Dec-2016 by the IAEA to LSVEC customers suggests the $\delta^{13}$C-uncertainty of LSVEC at ±0.15 ‰; this value is also used by A.Schimmelmann et al. 2016 (see http://pubs.acs.org/doi/abs/10.1021/acs.analchem.5b04392). The present interpretation of the message distributed by the IAEA is misleading.

All in all I find the uncertainty evaluation presented in the manuscript as unclear, confusing and partly misleading. The uncertainty evaluation for δ$^2$H may suffer for similar reasons. Given the problem with LSVEC, the $\delta^{13}$C uncertainty presently achieved appears not fulfilling the requirements.

Sergey ASSONOV (reference material specialist for stable isotopes)
IAEA Environment Laboratories, IAEA

---

## Author Comment (AC1) · 10 Jun 2016

**Author's Comments to the reviewers**

First of all, we would like to thank both reviewers and the author of the additional comment for their insightful and constructive critiques. As suggested by both reviewers, we have considerably re-written and re-structured large parts of the manuscript. Therefore, page and line numbers of the specific comments by the reviewers will not match the numbers in the revised manuscript. The revised manuscript considers every point mentioned by the reviewers and the comment, unless specifically addressed otherwise. We dedicated specific emphasis on the following aspects:

1) In the introduction, we present a review on the history of produced scale anchors at a range of laboratories analysing $\delta^{13}$C-CH$_4$ and $\delta^2$H-CH$_4$. This list provides a good overview on the magnitude of potential inter-laboratory offsets and their variability over time. 2) We present a dedicated method section of the analytical setup that has been developed at MPI-BGC to analyse $\delta^{13}$C-CH$_4$ and $\delta^2$H-CH$_4$ in air samples. 3) We present a dedicated method section on the origin/development of the scale anchors at IMAU and MPI-BGC. 4) We have improved and clarified the applied terminology. For example we define the use of "calibration" in comparison to "measurement", we refer to the produced gas mixtures as synthetic CH$_4$-in-air standards and to the pure CH$_4$ gases as primary and secondary CH$_4$ gases. "Working standard" is abbreviated WS, "certified reference material" CRM, "reference material" RM and "matrix reference material" (e.g. CH$_4$ in air) as m-RM, complying with recommendations from IAEA TecDoc 1350. 5) We revised the calculation of the uncertainties and dedicated a separate method section to present our calculation method. All data presented in the manuscript include the uncertainties of the full traceability chain where possible (CRM➔WS➔primary CH$_4$➔secondary CH$_4$). The presented uncertainties include the most recent development in CRM's, i.e. the new uncertainty for LSVEC. 6) We revised the comparison between MPI-BGC and the previously published data/method of Sperlich et al., (2012). This includes a revision of the Sperlich et al., (2012) data and their uncertainties to include the full traceability chain. Moreover, we present new comparison experiments between the two methods of MPI-BGC and Sperlich et al., (2012) to discuss/support the methods presented in this manuscript. 7) The experiments to evaluate the potential for analytical errors of the new methods are explained and discussed in the main text in more detail, full details are provided in the Appendix.

Our response to the reviewers comments is indicated in blue in the following.

**Reviewer 1 (anonym)**

Comments to "Development and evaluation of a suite of isotope reference gases for methane in air" by Sperlich et al.

**Summary of manuscript**

The authors set up measurement systems for anchoring $\delta^{13}$C-CH$_4$ and $\delta^2$H-CH$_4$ to the international reference isotope scales. They also prepared a suite of isotope reference gases (pure CH$_4$ gases and synthetic CH$_4$-in-air gases) and calibrated them. The synthetic CH$_4$-in-air gases will be made available for intercomparison to achieve high compatibility in the atmospheric monitoring community.

**General comments**

This study serves a fundamental contribution to the research community in study of the global CH$_4$ cycle. Increasing number of data of CH$_4$ isotopic ratios in air is becoming available, especially since the development of continuous-flow measurement system. However, datasets from different laboratories cannot be currently merged due to un-negligible and unidentified inter-laboratory calibration offsets. This hampers optimized use of isotope data for better understanding of the CH$_4$ cycle. This study is a start-up of anchoring measurements of different laboratories. The authors set up the current best available measurement methods, systems and calibration strategies. Further intercomparison efforts to distribute the synthetic CH$_4$-in-air standard gases developed in this study to worldwide laboratories are also an important next step. This study has significance with good measurement quality, and the manuscript is fairly well organized; however, many parts are not written in a clear and concise way and do not contain complete information. In my opinion, this study should be definitely published but the current manuscript version is not suitable for publication. I realize that this manuscript will be an important long-term reference for the research community. This is why I think that the manuscript should meet high level of transparency for all detailed information and clearness in description to help following researchers. It is regrettable that number of my comments below includes points that could have been thoroughly considered and corrected with the authors' responsibility before submission. I would like to encourage the authors to complete all details and rewrite the whole manuscript to improve the readability, after which this manuscript might be evaluated again.

Thanks a lot, we hope we achieved this goal in the revised manuscript.

It is very important to unify terms throughout the manuscript (not to mix up different terms for same meaning).

Better defined throughout revised manuscript.

In particular, I suggest to clearly define hierarchy of the gases at early part of the manuscript and always use the terms defined.

OK.

Name of the gases are bracketed by "" (e.g. "Megan"), but I do not think this necessary.
Matter of taste, we left it as it was.

I also suggest modifying Figure 1 so that readers easily find which gas is at which level (exact correspondence between Figure 1 and Tables 1 and 2). You might consider a figure similar to the attached figure and get it merged into Fig. 1 (see also specific comments to Fig. 1). This would improve traceability.
OK.

To my understanding, the hierarchy is (1) IAEA "reference materials", (2) MPI-BGC "working standards" ($H_2O$ and carbonates), (3) "master" $CH_4$ gases (or "MPI-BGC primary" might be an alternative), (4) "secondary" $CH_4$ gases—these are also called "calibration" in tables, but I think "secondary" (or "MPI-BGC secondary") used at other places fits better, and (5) "synthetic $CH_4$-in-air standard" produced by dilution of the secondary. Since iSAAC system plays an important role, the authors should elaborate at least overview of the system (e.g. materials and temperature of preconcentration traps, separation columns, etc.). Is it similar to Brass and Röckmann (2010)? There are many references citable. The authors might briefly describe the system even if they consider a separate publication.
OK. Respective publication has just been published (Brand et al., 2016), nevertheless we dedicate an entire section to describe the measurement method in brief.

One more point on iSAAC—it seems that iSAAC is operated with "Carina-1" with nominal values on the IMAU scale, but this is not written explicitly. The authors should clearly describe this in the manuscript.
OK.

The IMAU scale is in the manuscript considered to be an established scale independent from the MPI-BGC scale developed in this study. Description of origin of the IMAU scale is also important for perfect peace out of the manuscript. I suppose that the IMAU scales (both $\delta^{13}C$ and $\delta^2H$) are also ultimately referenced to some of the IAEA reference materials. The authors should mention to this, not only cite to the technical paper Brass and Röckmann (2010). I understand that, as appeared in this study, during the course of the long history, systematic offsets could arise due to different measurement methods, different maintenance strategies etc. even that both scales are on the identical VPDB and VSMOW scales in theory—that is, the problem is in calibration at individual laboratories. It is important to describe this fact. This also helps future intercomparison planned.
OK.

Also confusing is that the gas "Carina-1" has $\delta^{13}C$ and $\delta^2H$ values on the IMAU scale. The authors might present those values with a note that they are on the IMAU scale, since they are listed in Table 2.
OK.

**Specific comments**

P2 L2: "Isotope ratios of trace gases" to "Isotope ratios of $CH_4$".
OK.

P2 L2: I do not think isotope ratios are still "an emerging tool"—they have tested over decades.
"a powerful tool"

P2 L3: Quay et al. (1991, 1999) are also important studies that showed quantitative use of isotopic compositions for understanding the global $CH_4$ cycle, which could fit this context.
OK.

P2 L16: "papers" to "the paper"
"papers that … (e.g. Levin 2012, Sapart 2013 and Schaefer 2016)."

P2 L17ff: The authors might list the IAEA reference materials used in the cited references.
OK.

Also, the authors might mention to the fact that most (or all?) calibrations in the cited references were linked to the IAEA certified materials and that in theory on identical scales. It would highlight the fact that calibration offsets have arisen during scale propagations in individual laboratories—this probably corresponds to more ambiguous word "the diversity of referencing trajectories" (P2 L21) in the manuscript.
We list CRM's used for scale anchoring by a range of laboratories and also include the revisions of their isotope values and their uncertainties. This is a technical but informative illustration of sources for inter-laboratory offsets. Errors due to propagation add on top of this.

P2 L21: First, here in the context the authors should refer only to papers already published and thus "W.A Brand pers. comm." should be left out; second, personal communication with one of contributing authors is strange—same for other places. An alternative might be "W. A. Brand, unpublished data" if this matches the context. Otherwise unpublished results should not be cited unless clear necessity—it would not help readers and traceability of the manuscript.
OK.

P2 L22: "referencing" to "calibration"
OK.

P2 L23: I suggest to start this paragraph with a phrase like "For isotope ratios of $CO_2$ in air, …", so that readers better understand that the authors intend to introduce an analogical method already applied for another gas.
OK.

P3 L7ff: Please unify use of "the Principle of Identical Treatment" and "PIT" everywhere in the manuscript.
OK.

Figure 1: As pointed out in the general comments, the authors might enrich this figure by adding information on which standards in Table 1 are at which level in this figure. Another idea is to split this figure both for $\delta^2$H and $\delta^{13}$C as the figure below. Another good reference is Figure 2 of Sperlich et al. (2012).

[Figure]

OK.

P3 L15: I do not think that "commercial provider", "methanogenic origin" and "isotopic origins" should be listed in parallel with equal stress. Differences in methanogenic origin and isotopic composition in $CH_4$ is highly linked; the former is a primary factor for the latter. Roughly speaking, commercial providers do not always care about it.
OK.

P2 L17: "and are therefore of known isotopic composition" should be left out. The "Biogenic" and "Fossil" gases are treated as unknown in this study and removing the phrase would reduce risk of misunderstanding.
OK.

Table 1: I am not sure if "create" is a suitable word. I think "produced" sounds more common.
OK.

P4 L7: The contents in the first paragraph of section 3.1 should be moved here.
OK.

P4 L15: "We show the most recent values…" The table gives references, but the caption mentions only to Brand et al. (2014)—this is confusing. If original papers are cited in the table, I would delete this sentence of the caption.
OK.

P5 L4: The abbreviation "IRMS" appears here for the first time.
OK.

P5 L6: "after the conversion to $H_2$"—since there is "to covert $CH_4$ to $H_2$" just a one line above, this phrase is a redundancy. This sentence may be like "…to convert $CH_4$ to $H_2$ (+carbon) for subsequent measurement of $\delta_2H$-$CH_4$ in pure $CH_4$ gases on IRMS."
Paragraph re-formulated.

P5 L4–L9: In my first reading, I (probably) misunderstood that $CH_4$ and $H_2O$ in sample are converted to $H_2$ in different reactors. Besides the above suggestions, I suggest possible reorganization of the sentences: "We use a TC/EA coupled to an IRMS via an open split for $\delta^2H$ measurements of $CH_4$ and water; $CH_4$ in sample gas is introduced into the TC/EA and converted to $H_2$ in a glassy carbon reactor maintained at 1450°C; water sample is injected through a heated septum at 130°C into the same reactor and converted to CO and $H_2$; the converted $H_2$ is measured on the IRMS for $\delta^2H$."
Paragraph re-formulated.

P5 L9: "hydrogen isotopic composition" to "$\delta^2H$"
P5 L11: "that is configured as shown in Fig. 2" to "configured as Fig. 2"
Matter of taste, we prefer original wording as the plumbing of a Figure cannot be configured but the plumbing that is shown in a figure.

P5 L12: "Typical $CH_4$ feed flow rates range between 2-3 mL/minute." to "Typical $CH_4$ feed flow rate is 2–3 mL/minute." How much volume of $CH_4$ is actually injected?
"Typical flow rates of $CH_4$ range between 2-3 mL/minute."

P5 L14: Leave out "now". My suggestion is like: "Measurement sequences are configured by combining injections of $CH_4$ gas and reference water materials into the reactor, which are via the 10-port valve (Fig. 2) and septum from an autosampler, respectively." I would leave out ", where both $CH_4$ and $H_2O$ are converted…"
"While $CH_4$ gases are injected manually, $H_2O$ is introduced via autosampler."

P5 L16: I would leave out "Apart from the injection procedure…" because it comes up later soon again.
OK.

P5 L18–19: "The amounts" to "Amounts". "to achieve matching peak…" to "to match peak…". "during IRMS analysis" to "of the IRMS output signal". But this sentence might be left out because same (and more in-detail) explanation appears page 7.
OK.

P5 L20: "Fig. 4" to "Fig. 3"
OK.

P6 L8: "This configuration enables alternative injections of $CH_4$ gases with known and unknown $\delta^2H$ values in an identical fashion; the calibrated "Megan" and "Merlin" served as known reference gases for calibrations of the other $CH_4$ gases (Table 1)."
"For the calibrations of primary $CH_4$ gases, the two sample loops are fed by the same $CH_4$ gas (connecting vent 1 and $CH_4$ port 2). The sample loops are fed by two different gases for the calibration of secondary against primary $CH_4$ gases (Table 1)."

I do not get what "the respective isotope scales" means.
Removed.

P6 L14: "applying" to "employing"
Section re-formulated.

P6 L15: "working reference waters" to "working standard" because they seem to be abbreviated as "ws" in Table 2. Decapitalize www-"J"1 and BGP-"J"1 to harmonize with Table 2.
OK.

P6 L20ff: use "PIT" throughout the manuscript if this term is abbreviated.
OK.

P7 L4: leave out "several", otherwise write more explicitly.
Re-formulated.

P7 L8: "assumed to be" kept constant
OK.

P7 L13: "…, which guarantees same level of $H_3$-factor correction and allows use of the standard integration software (ISODAT, Thermo, *Company name & place as other places*)"
"The similarity between $CH_4$-derived and the $H_2O$-derived $H_2$ peaks allows the use of the standard integration software (ISODAT, Thermo Finnigan, Bremen, Germany)."

P7 L14–18: "We performed…" Please write explicitly about what were tested. I do not understand which experiments in appendix correspond to these descriptions. I could not find descriptions that support these sentences—it sounds like that the authors argue good performance of their measurements without showing any clear evidences.
OK.
1. memory
2. septum temperature
3. reactor temperature for quantitative conversion of both $H_2O$ and $CH_4$

P9 L5: "routinely used"—does this mean the system routinely used at MPI-BGC or at other laboratories? This sentence may intend to endorse reliability of measurements in this study, but it would not work if the authors mean the latter. Even if the former, just "routinely used" cannot justify. I would write more explicitly—for instance, "Similar systems were used for… in previous studies (…et al.)" or "MPI-BGC has operated this system for… over X years (…et al.)." The same comment is also for P5 L6.
P5 L6: OK. We strongly disagree with the speculation regarding our intention to choose the formulation. The cited references were published almost 20 years ago (1999), TC/EA-IRMS for $H_2O$ analysis is a standard method. The use of a well-established method for the production of reference materials is required by the IAEA (TecDoc 1350). We mention this explicitly in the revised manuscript to prevent from speculations regarding the intention of the formulation.

P9 L5: Same argument. EA-IRMS are workhorses in numerous- laboratories all around the world. Listing more previous studies than the two given references seems to exceed the norm on a 2-decade old technique. New is our modification that allows to use this system for gas samples.

P9 L7: My suggestion: "Outflow of the 10-port valve enters into a combustion furnace with helium carrier gas stream of 10 mL/min through an 1/16 inch tubing (70/30% Cu/Ni alloy) specially fitted to the EA system (Fig. 4)." How important is "the oxygen plume region"? If this is critical, elaborate more. Otherwise the following sentence may be enough.
"We fitted a 1/16" tube of 70/30 % Cu/Ni alloy to the EA and used the previously described 10-port valve to inject the CH4 samples into the EA with a 10 mL/min helium flow (Fig. 4)."

Figure 4: "oxidation"—the text use different terms like "combustion chamber" and "combustion reactor". This confuses readers. "combustion furnace" (also "reduction furnace" might be good.
"Figure 4: 10-port valve for manual $CH_4$ injections coupled to the EA-IRMS system through custom made gas inlet into combustion (oxidation) unit. The proportions of illustrated components is chosen to increase clarity."

P9 L13: "the combustion reactor"—is this different from "the combustion chamber" in the preceding sentence? Use one term if not.
OK.

P9 L14: "All samples are oxidized to $CO_2$ in the combustion furnace maintained at 1020° C (Werner et al., 1999) and experience identical analytical treatment (PIT) thereafter." I think that the authors monitor furnace's temperature but not temperature at which sample actually reacts. Do not mix up "combustion" and "oxidation" so often, otherwise readers wonder if the authors intend to use them with different meanings.
OK.

P9 L16: "The sample is dried by passing through a Nafion dryer (…) and a trap filled with $Mg(ClO_4)_2$, and then introduced into a GC column (3 m×1/4 inch Porapak PQS, CE instruments) held at 80° C."
"The sample is dried by passing through a NafionTM membrane (Perma Pure LLC, Toms River, NJ, USA, not shown in Fig 4) and a $Mg(ClO_4)_2$ trap before it enters the GC column (3m, 1/4", Porapak PQS, CE instruments) held at 80°C."

P10 L1–3: Circumlocution. My suggestion: "By alternating injection of $CH_4$ gas and carbonate reference materials such as LSVEC, Mar-j1 and ali-j3, we referenced our Master CH4 gas (Megan and Melrin) to the VPDB isotope scale."
Rephrased.

Later NBS-19 also appears (not listed in Table 2), and the authors describe Megan and Merlin were calibrated against NBS 19 and LSVEC. How were Mar-j1 and ali-j3 used? Were they used to assign values to the Master gases? Or just for measurement control? Clarify this.
OK.

For $\delta^2$H calibrations, the authors used "in house standards" to assign values to the Master gases, but here for $\delta^{13}$C, the Master gases are directly measured against the IAEA materials.
That is correct for $\delta^2$H. $\delta^{13}$C calibrations were made against LSVEC directly and one internal working standard (MAR-J1 in most cases and ALI-J1 once).

I do not understand what "over three independent periods each" means.
""Megan" and "Merlin" were each calibrated on three different days to determine the external reproducibility of the $\delta^{13}$C results."

P10 L5: "analysis" to "analyses"
OK.

P10 L13–P11 L1: "The first three square-shaped peaks and the last tailing peaks are for pure $CO_2$ working gas and $CH_4$- and $Li_2CO_3$-derived $CO_2$, respectively."
Matter of taste.

P11 L2: "…(with peak widths of 101 s and…)
OK.

P11 L7: "the two-step calibration strategy approach"—Clarify meaning of this. Does it mean the calibration has two anchoring points on the VPDB (VSMOW) scale?
Due to the concerns expressed in all reviews and comments, the entire section 2.4 Measurement uncertainty and error propagation is now re-formulated.

P11 L8: As mentioned earlier, NBS 19 appears here for the first time and not listed in Table 2. Megan and Merlin are bracketed by "" at other places, but not here.
OK.

P11 L14: "…has been found, which has been…"
OK.

P11 L18: "BGC ISOLAB"—Would not "MPI-BGC" work as other places?
OK.

P11 L27: "The secondary or "calibration" $CH_4$ gases"—as mentioned earlier, define the hierarchy of standards early in the manuscript so that confusing words like here do not come up.
OK.

P11 L30: Which is correct "Master $CH_4$ gas" or "master methane gas"?
Now primary $CH_4$.

P12 L1ff: I suggest to use a term "synthetic CH$_4$-in-air standard" for gases produced by dilution of the secondary CH$_4$ gases.
OK. Note we produce synthetic CH$_4$-in-air standards from both primary and secondary CH$_4$

P12 L2: Same comments as P11 L18.
OK.

P12 L3: "This system (named ARAMIS) is used to produce synthetic standard gases with atmospheric CH$_4$ mole fraction levels."
"The MPI-BGC operates an analytical system (named ARAMIS) to dilute pure CO$_2$ with CO$_2$-free air to atmospheric CO$_2$ mole fraction without isotopic fractionation (Ghosh et al., 2005). We use ARAMIS to dilute an aliquot of primary or secondary CH$_4$ with CH$_4$-free air to atmospheric CH$_4$ mole fractions (~2 ppm) in 5-L glass flasks with a final filling pressure of 1.8 bar absolute."

P12 L4: "We diluted aliquot of the secondary CH$_4$ gases with CH$_4$-free air to…"
We diluted and compared primary and secondary CH$_4$ gases (Table 6).

P12 L5–6: "The CH$_4$-free matrix air, which was produced by target-mixing ultra-pure constituents, contains N$_2$, O$_2$, N$_2$O and Kr at atmospheric levels, so that composition of the produced gas is as close to ambient air as possible."
OK, see next comment below.

P12 L7–8: "…to account for the interference effect on the $\delta^{13}$C-CH$_4$ measurements using GC-IRMS systems (Schmitt et al., 2013)." I would leave out "so that…" but add this part to the preceding sentence.
"The CH$_4$-free matrix air has been target-mixed from ultra-pure constituents and contains N$_2$, O$_2$, N$_2$O and Kr at atmospheric levels, so that the composition of the produced CH$_4$-in-air standards is as close to ambient air as possible. Krypton was added to this matrix air to account for the measurement artefact during GC-IRMS analysis of CH$_4$ for $\delta^{13}$C (Schmitt et al., 2013)."

P12 L8: "A sensitive analysis…" I do not understand what this sentence means. An upper limit for what? Sensitive to what? What is high-precision gas-chromatography? Is it different from GC-IRMS?
"A blank analysis of the CH$_4$-free-air yielded a maximum CH$_4$ blank of 0.5 ppb."

P12 L9: I do not think Table 1 gives "further details."
OK, sentence removed.

P12 L11: "an average standard deviation" to "average standard deviations", but what does an average standard deviation mean? Standard deviations for measurements of synthetic standard from multiple dilutions were averaged?
OK, sentence removed.

P12 L12: Personal communication with one of contributing authors is odd. For this sentence, the authors should give complete description of measurement precisions of the analytical system in section 2.6.

OK.

P12 L17: Same comment as P12 L12.

OK.

P12 L16–L20: As described in general comments, the authors should describe at least overview of the iSAAC system. Since it seems to be a continuous-flow system, likely similar to Brass and Röckmann (2010), the authors should give configuration of the system with basic information of key components, so that at least relevant researchers can understand the system well, because this system is a key to calibrate the synthetic $CH_4$-in-air standards.

OK.

P12 L22: As described in the general comments, even that readers are led to Brass and Röckmann (2010) for very details, the authors should describe basics of the IMAU scale. That is, with what types of laboratory standards they maintain long-term consistency of their scales, how and to what IAEA materials their scales are ultimately referenced. As I mentioned earlier, their scales are also the VSMOW and VPDB scales; the IMAU scales are identical to the MPI-BGC scales in theory.

OK.

P12 L23: ""Carina-1" as master reference gas for the iSAAC system" means, as long as any gases are measured by iSAAC system, they are assigned values on the IMAU scale? This should be clarified and it would help understand Table 4 and relevant texts.

OK, section re-worded and re-structured.

P12 L24–25: Are Carina-1 and Carina-2 of identical origin and should they agree both $\delta^{13}C$ and $\delta^2H$ in theory? This is unclear. Only the description "Jena air" in Table 1 does not guarantee it. Regarding the offset in $\delta^2H$, was the cause identified? Also, the authors might present $\delta^{13}C$ and $\delta^2H$ values of Carina gases on the IMAU scales.

OK, section re-worded and re-structured.

P12 L26: "The synthetic isotope reference gases" to "The synthetic $CH_4$-in-air gases",

OK, section re-worded and re-structured.

"previously calibrated $CH_4$ gases" to "the secondary $CH_4$ gases"

OK this is clarified in the revised version.

P12 L27: "The results of these measurements are compared to the calibration results of the secondary $CH_4$ gases so that the differences between the calibrations in this study and the iSAAC measurements against Carina-1 indicate the offsets between the MPI-BGC and IMAU scales."

OK, section re-worded and re-structured.

P13 L4: "carbon and hydrogen isotope ratios" to "$\delta^{13}$C-CH$_4$ and $\delta^2$H-CH$_4$"
OK.

P13 L7: "in the reference hierarchy"; "calibration CH$_4$ gas" to "secondary CH$_4$ gas"
OK.

P13 L8: "Therefore" to "As a result"
OK.

P13 L8: "primary reference materials" to "international reference materials"; or the authors might use "RM" for the IAEA certified reference materials, with this stated in early part of the manuscript
OK., unified abbreviations throughout revised manuscript.

P13 L9: "against a Master-CH$_4$ gas Megan"
OK.

P13 L11: "Both Megan and Merlin are fossil in origin with typical $\delta^{13}$C-CH$_4$ and $\delta^2$H-CH$_4$ signatures (e.g. …). The two Master-CH4 gases are similar in $\delta^2$H-CH$_4$ with calibrated values of −168.0±0.6‰ for Megan and −165.7±0.6‰ for Merlin. The calibrated $\delta^{13}$C-CH$_4$ values are −40.75±0.07‰ for Megan and −39.07±0.07‰ for Merlin." Megan's $\delta^2$H is −168.1 in Table 3, but −168.0 here in the text.
OK, section re-worded and partially shifted into section 2.1 as suggested above.

P14 L5: I would leave out lines "Fossil comparison" and "Biogenic comparison" from the table and bring them to footnote of the table. This might improve ease of viewing of the table by focusing only on calibration result of this study.
OK. Revised manuscript has dedicated section for comparison between CIC and MPI-BGC.

P14 L7: "were then mixed with $\delta^2$H-spike gas to produce Martha-2 and Mike-2 with $\delta^2$H-CH$_4$ values higher than or similar to that of the tropospheric CH$_4$, respectively."
OK.

P14 L11: "Results for secondary CH$_4$ gas calibrations against Master CH$_4$ gases"
OK, "Results for secondary CH$_4$ gas calibrations against primary CH$_4$ gases".

P14 L13: "…spiked CH$_4$ mixtures, and thus cover wide range of $\delta^{13}$C-CH$_4$ (…) and $\delta^2$H-CH$_4$ (…)." I would leave out ", which include…"
OK.

P14 L15: "create" to "produce"
OK.

P14 L15: "a CH$_4$ gas with $\delta^2$H-CH$_4$ close to that of the tropospheric value, …"
OK, paragraph added to section 2.1 and re-worded.

P14 L16: "a fossil CH$_4$ gas"—Is this the "Fossil" gas? I wonder if "diluted" is the correct word, because dilution usually means lowering mixing ratio of certain compounds, but here the CH$_4$ gases are almost pure gases and mixture of such gases just result in no change in CH$_4$ mixing ratio.

OK, paragraph added to section 2.1 and re-worded. (diluting referred to deuterium content but that was bad choice of wording).

P15 L1: "Results for calibrations of synthetic CH$_4$-in-air standards"
OK.

P15 L2: "Aliquots of the secondary CH$_4$ gases were diluted with CH$_4$-free air to produce the synthetic CH$_4$-in-air standards (section 2.5) for analysis on the iSAAC system (section 2.6)." Large part of L3 is redundancy.
OK.

P15 L4: "the diluted CH$_4$ reference gases" to "the synthetic CH$_4$-in-air gases"
OK.

P15 L4: "…against Carina-1 on the IMAU scale."
OK.

P15 L5–7: My suggestion: "We calculate the difference between the calibrations of the secondary CH$_4$ gases on our measurement systems (sections 2.2 and 2.3) and the synthetic CH$_4$-in-air standards on iSAAC, $\delta iSAAC - \delta sec$ (Table 4); the value indicates calibration offsets between the MPI-BGC and IMAU scales, if we assume no isotopic fractionation in the dilution process."
Sentence now in section 2.8 "We calculate the isotopic difference ($\delta_{iSAAC} - \delta_{pure}$) between the measurements on iSAAC and the calibrations of the pure CH$_4$ gases (Sect. 2.2 and 2.3), which indicates the correction to anchor the measurements at MPI-BGC to JRAS-M16."

P15 L8–9: My suggestion: "Our experiments show a good agreement for $\delta^{13}$C-CH$_4$ with an average difference of +0.02±0.08‰, but a significant systematic offset of +4.0±1.1‰ for $\delta^2$H-CH$_4$." What is the cause of the $\delta^2$H-CH$_4$ offset? The authors should discuss on possible sources of the offset here or in the next section.
OK.

P15 L13: "the pure CH$_4$ gas" to "the secondary CH$_4$ gas"
The entire discussion on the offset in "Biogenic" is re-worded.

P15 L13: "A sudden drift" by what kind of reason?
Statement was too speculative and is removed from the revised manuscript.

P15 L19: "Comparison of the calibrations on the new MPI-BGC scale developed in this study and iSAAC measurements on the IMAU scale (Brass and Röckmann, 2010)."
Table Caption re-worded to "Table 6: Differences in $\delta^2$H-CH$_4$ and $\delta^{13}$C-CH$_4$ between primary/secondary CH$_4$ gas calibrations and iSAAC measurements of the synthetic CH$_4$-in-air standards using the scale anchor based on "Carina-1". Differences are calculated as $\delta_{iSAAC} -$

$\delta_{pure}$. The bottom line shows the average and the standard deviation ($1\sigma$) of considered differences, excluding the value of "Biogenic" (°) as described in main text."

P15 L21: "the name of the Master/secondary $CH_4$ gas that was diluted to the synthetic $CH_4$-in-air standard for iSAAC measurements"
See above comment.

P15 L23: "the mean difference" to "the average difference"—use one term both for caption and table.
See above comment.

P15 L24: "Therefore, this value (marked with °) was excluded for calculation of the average scale difference."
See above comment.

P16 L2: "…to calibrate the pure CH4 gases for…"
OK.

P16 L3: "methane" to "$CH_4$"; "…into the isotope measurement system that also analyses water and carbonate reference materials, and thus subject to PIT."
"Samples and reference materials were always analysed in the same analytical systems, thereby complying with the PIT as much as possible."

P16 L5: "The online oxidation of $CH_4$ to $CO_2$ (and $H_2O$) is considered to produce no isotopic fractionation." This sentence needs references.
Sentence removed.

P16 L6: "However, $CH_4$ is relatively stable chemically; complete oxidation of $CH_4$ thus requires high temperature and surplus of oxygen."
Sentence re-worded to: "Quantitative oxidation of $CH_4$ during $\delta^{13}$C-$CH_4$ analysis requires high reaction temperatures (e.g. Dumke 1989)".

P16 L7: "allow for" to "leave"
Sentence re-worded to: "$CH_4$ is a potent source of protonation in the IRMS ion source (Anicich, 1993)".

P16 L8: "$CO_2$ present" to "presence of $CO_2$"
Section re-worded.

P16 L9: "This source of analytical error" to "This effect"
Section re-worded.

P16 L10: "don't" to "do not"
Section re-worded.

P16 L11: "In the MPI-BGC systems" to "In the MPI-BGC EA-IRMS system"; Is this also the case for iSAAC? Clarify.
Section re-worded.

P16 L12: "methane" to "CH$_4$" (2 places)
OK, for all of manuscript.

P16 L14: "quantitative" Does it mean combustion efficiency of 100% or close to 100% without measureable isotopic fractionation? Write explicitly. Meaning of "quantitative" is unclear also for many other places.
Details provided.

P16 L15: "quantitatively"—same comment as the above. I would write: "It has been demonstrated that introduction of carbonates into high-temperature combustion furnace yields CO$_2$ conversion resulting in high-precision $\delta^{13}C$ measurements (…)."
OK.

P16 L16: "…oxygen isotope composition is altered completely in the conversion process from the original carbonate to the product CO$_2$."
Section re-worded.

P16 L17: I do not understand what this sentence means.
Section re-worded.

P16 L18: "ambiguity" to "uncertainty";
OK.

"extracting" to "calculating";
OK.

"values" to "value";
We leave "values" and use plural in "ion currents" thereafter.

"tends to cancel" means that it is not 100% guaranteed and there are exceptions. The authors should better justify. Besides, this paragraph seems to be readable for only expert readers who can easily refer to equations for $^{17}O$ correction. The authors should present "kind" introduction at the beginning of this paragraph on why this matters—I think this needed for AMT which expect readers more general than e.g. RCM.
Section re-worded.

P16 L21: "hydrogen" to "$\delta^2H$ measurement"
"…$\delta^2H$ analyses,…"

P16 L22: "quantitative conversion"—same comment as the above.
"…to produce high conversion yields of …"

P16 L23: "methane" to "CH$_4$";
OK for all.

My suggestion: "Major artifact can arise from more variable surface adhesion of H$_2$O than CH$_4$ in the combustion furnace before they are converted to H$_2$ (and CO/carbon)."
The suggested sentence is misleading. We discuss this in detail as this highlights the importance of the information in Appendix A, which might otherwise be under-appreciated: *i*) The surface adhesion effects of H$_2$O are not variable at constant reactor temperatures and constant H$_2$O amounts, in fact they are a reproducible function of T, which is why this system works for the calibration of H$_2$O against H$_2$O. This is demonstrated in Appendix A, Figure A2, where the variability in $\delta^2$H-H$_2$O is clearly reactor temperature dependent but reproducible for every temperature. Note this temperature effect is insignificant for CH$_4$, as CH$_4$ shows no significant adhesion to surfaces of the analytical system. *ii*) Significant adhesion effects of H$_2$O occurs at several places, inside the reactor and on the septum, as discussed in Appendix A. *iii*) This paragraph is not about the combustion reactor but about the pyrolysis occurring in the classy carbon reactor of the TC/EA-IRMS system at temperatures much higher than combustion temperatures and in the absence of oxygen. *iv*) The artefacts range from very strong in case of reactor temperature (~15 ‰) to significant in the case of septum temperature (~3 ‰) over the tested temperature ranges, which is why we use the term "possible artefacts".

We change this sentence in the revised version to "Possible artefacts can arise mainly from the stronger surface activities of H2O versus CH4 prior to the conversion to H2 (and CO or carbon).".

P16 L24: "water" to "H$_2$O";
OK for all places.

I would write: "This can lead to memory effect in the $\delta$2H-H2O measurements, then corrections or discarding initial injections are needed (…)"
The suggested sentence is misleading. In fact, the memory effect is induced by H$_2$O adhesion to the surfaces of the analytical system (reactor, septum). The artefact due to system memory between injections of the same H$_2$O sample decreases with every injection until it is insignificantly small. If we inject a sequence of H$_2$O samples, followed by a sequence of CH$_4$ samples, then the first $\delta^2$H-CH$_4$ analysis may in some cases be affected by the memory effect caused by H$_2$O adhesion due to desorbing H$_2$O. However, that was not consistent.

We change this sentence in the revised version to "H$_2$O injections can lead to memory effects, which need to be taken into account in $\delta^2$H-H$_2$O and subsequent $\delta^2$H-CH$_4$ analyses, either by discarding initial injections or by correcting for it (Werner and Brand, 2001)".

P16 L26: "In addition, we found a minor dependence of…";
OK.

"In the appendix" to "In Appendix A"
OK.

P16 L29: "with a large number of analyses"—number of analyses is given neither in Table 3 nor text. Without this, this description is not justified. I would write instead: "We have presented calibration results of the secondary CH4 gases Fossil and Biogenic (Table 3). Both gases…"
This paragraph is removed from this section in the revised manuscript.

P16 L30: Leave out "in an earlier study"; "the" to "a"; "combusting" to "combustion of"
This paragraph is removed from this section in the revised manuscript.

P16 L31: "methane" to "$CH_4$"; "sampling of";
This paragraph is removed from this section in the revised manuscript.

"consecutive" to "subsequent"?
This paragraph is removed from this section in the revised manuscript.

P16 L31–33: "Sperlich et al. (2012) analyzed the $CH_4$ derived $CO_2$ for $\delta^{13}C$-$CH_4$ on a dual inlet IRMS and the $CH_4$ derived $H_2O$ for $\delta^2H$-$CH_4$ on a TC/EA-IRMS system similar to this study or cavity-ring-down spectroscopy."
OK, sentence is moved into method Section 2.9.

P16 L33: My suggestion: "Our calibration results are in overall agreement in both $\delta^{13}C$-$CH_4$ and $\delta^2H$-$CH_4$ with the previous values by Sperlich et al. (2012) within the uncertainties of both measurements (Table 3)."
Section re-worded.

P17 L1–3: "However, our calibration results for Biogenic and Fossil appear to…, suggesting systematic offsets."
Section re-worded.

P17 L3: This statement is strange. After this, the authors argue that calibrations by Sperlich et al. (2012) are less robust (which I do not think justified), but here the authors argue that their measurements are supported by agreement with the unreliable measurements by Sperlich et al. (2012).
Section re-worded and further developed.

P17 L5: "a large number of measurements"—same comments as P16 L29. I do not think that long-term use itself guarantees accuracy and robustness of a measurement system. Long-term use with unidentified artifacts can happen. The Kr interference on $\delta^{13}C$-CH4 measurements is a good example—GC-IRMS had used for more than a decade until it was found.
See comment above.

P17 L6–8: I do not think that the statement in these sentences is justified. Sperlich et al. (2012) indeed has limited number of measurements, but did thorough treatments for complete combustion and reduction. Therefore, combustion and reduction by Sperlich et al. (2012) might be more complete than the online conversions made in this study. If so, the

calibrations by Sperlich et al. (2012) might be more robust even if number of measurements is less. To keep the authors' argument, the authors should describe weak points of Sperlich et al. (2012) specifically.
See comment above.

P17 L11: "uncertainty" to "uncertainties"
See comment above.

P17 L12: "an indicator"; "create" to "cause"
See comment above.

P17 L14: I do not understand what the authors argue here. With "CH$_4$ reference materials" (which I do not understand what the authors refer to), what would the author do for further tests? What is the authors' best idea?
See comment above.

P17 L15–17: "The total propagated uncertainties in our calibrations are smaller than or similar to uncertainties of widely used analytical systems for $\delta^{13}$C-CH$_4$ and $\delta^2$H-CH$_4$ in air/ice core samples (references are needed). Therefore, a suite of standard gases developed in this study can help to increase the compatibility between international laboratories."
This paragraph is removed from this section in the revised manuscript, as this is not the case when the uncertainty of LSVEC is considered.

P17 L21: "The number" to "Number"
?

P17 L22: "could lead to" to "provides";
"…and combining data from multiple laboratories could enable new science and increasingly powerful analysis.".

delete "of the combined data sets"
OK.

P17 L22–23: My suggestion: "However, such merged dataset has not been achieved by the lack of reference materials that enable direct intercomparison in the community."
"However, merging data from multiple laboratories for analysis is currently hampered by the lack of reference materials that enable the community to produce a unified data set."

P17 L25–26: The paragraph can just follow the previous one without line break;
OK.

My suggestion: "To deal with this problem, we prepared 12 pure CH$_4$ gases (the secondary CH$_4$ gases) and accurately referenced them to the international isotope scales VSMOW and VPDB for $\delta^2$H-CH$_4$ and $\delta^{13}$C-CH$_4$, respectively. These secondary CH$_4$ gases then were diluted to produce 8 synthetic CH$_4$-in-air standards in 5-L glass flasks."
"To overcome this problem and to improve compatibility between laboratories, we produced synthetic CH$_4$-in-air standards (JRAS-M16)."

P17 L26–28: I do not understand what the authors argue here. The authors say the synthetic $CH_4$-in-air standards were "tested for their use", but where in the manuscript did they evaluated usability of the gases? Section 3.3 does not seem to describe this. "separately" to "indendently".
Section reformulated in revised manuscript.

P17 L29: "synthetic atmospheric reference gases" to "synthetic $CH_4$-in-air standards"; "isotopic composition of $CH_4$" to "$\delta^{13}C$-$CH_4$ and $\delta^2H$-$CH_4$"
OK.

P17 L30: My suggestion: "These synthetic $CH_4$-in-air standards will help worldwide laboratories to anchor their measurement datasets to unified $\delta^{13}C$-$CH_4$ and $\delta^2H$-$CH_4$ scales shared in the atmospheric monitoring community, enabling compatible isotope ratio datasets for better understanding of the global $CH_4$ cycle."
Section reformulated in revised manuscript.

P18 L8: This appendix describes "preliminary experiments", but the authors state in the main text that they optimized the measurement condition.
"Experiments to enhance the performance of the analytical system for the calibration of $\delta^2H$-$CH_4$ with $H_2O$".

P18 L11: "This effect is avoidable by repetitive…"
We disagree. The effect is still there, but it is overwritten or overcome by repeated injections.

P18 L13: "However" to "Moreover"
OK.

P18 L14: delete "furthermore";
OK.

"scales with isotopic difference" to "depends on difference of isotope ratio"
Matter of taste, we prefer the original version.

P18 L15: delete "only"
OK.

P18 L16: I do not understand how this sentence is liked to the preceding sentence by "Therefore".
Section reformulated in revised manuscript.

P18 L17: delete "for" after "corrected"
We think "corrected for" is correct.

P18 L18–19: "…, as our system"—as I mentioned for P17 L5, long-term operation itself does not guarantee correctness, so the last sentence of this paragraph is not justified.

While we accept the argument for the Kr example (reviewer's comment to previous P17 L5), we disagree in this case as both cases are not comparable. The Kr effect was possible due to the lack of *i*) knowledge and *ii*) suitable reference materials. In the case discussed here, knowledge, suitable reference materials and experimental methods are available to guarantee sufficient control on system memory during $\delta^2$H-H$_2$O analysis.

P18 L21: "We made 106 injections of an identical H$_2$O samples…"

"We injected 106 identical H$_2$O samples".

P18 L22: "Septum" to "septum";

OK.

I would delete "however, there seems to…"

OK.

P18 L23: My suggestion: "A systematic increase of $\delta^2$H-H$_2$O with the septum temperature is apparent above 90° C, but the $\delta^2$H-H$_2$O value reaches the plateau around 130° C."

"A systematic increase of $\delta^2$H-H$_2$O with septum temperature is apparent above 90°C until $\delta^2$H-H$_2$O values plateau at septum temperatures around 130°C".

P18 L24: "At three highest temperatures"—The authors say the $\delta^2$H-H$_2$O value stabilized above 130°C, then the average should be calculated from the data above 130°C.

We removed the sentence as we do not have values for 130°C exact.

P18 L25: "The $\delta^2$H-H$_2$O values stabilized above 130° C suggests quantitative adequate conversion of H$_2$O processing without …"

Matter of taste, we prefer the original version.

P18 L26: "at the lower temperature range" to "below 90°C";

OK.

but my suggestion is for instance: "In contrast, the $\delta^2$H-H$_2$O values below 90°C show an insignificant slight increase with the septum temperature, which deviates from the pattern above 90°C." I do not understand the original sentence. What does "an offset" mean?

OK.

P18 L28–30: I do not understand these sentences. Elaborate better.

OK.

P19 L3: "fall onto a polynomial fit" is the authors' interpretation. Here the authors should write, for instance, as "the black line is the quadratic polynomial fit to the data above 90°C".
OK.

P19 L4: What is "the offset"?
Sentence re-worded.

P19 L6: The reasoning of taking a value from the polynomial fit is unclear.
Reformulated: "The error bars show 1σ standard deviations and the grey-dashed lines indicate the typical precision limit of 1 ‰ for $\delta^2$H-$H_2O$ analysis (Gehre et al., 2004) around the $\delta^2$H-$H_2O$ value of the polynomial fit for the septum temperature of 130°C (set point during calibration experiments). The grey dashed lines show that our $\delta^2$H-$H_2O$ analyses remain within a typical precision level as long as the septum temperature is controlled to ~130±10°C."

P19 L10: "high temperature" to "high-temperature";
OK.

"utmost" to "particular"
Matter of taste, we prefer the original version.

P19 L12: I would leave out "The temperature…"
? "The reactor temperature is…"

P19 L13: "…at different reactor temperatures (Fig. A2)."
OK.

P19 L16: "150 K" to "150°C"
OK.

P20 L14ff: I do not find where this appendix fits in the main text and what it supports.

We included the experiments of Appendix B into the discussion part of the main text.

---

## Author Comment (AC2) · 10 Jun 2016

**Author's Comments to the reviewers**

First of all, we would like to thank both reviewers and the author of the additional comment for their insightful and constructive critiques. As suggested by both reviewers, we have considerably re-written and re-structured large parts of the manuscript. Therefore, page and line numbers of the specific comments by the reviewers will not match the numbers in the revised manuscript. The revised manuscript considers every point mentioned by the reviewers and the comment, unless specifically addressed otherwise. We dedicated specific emphasis on the following aspects:

1) In the introduction, we present a review on the history of produced scale anchors at a range of laboratories analysing $\delta^{13}$C-CH$_4$ and $\delta^2$H-CH$_4$. This list provides a good overview on the magnitude of potential inter-laboratory offsets and their variability over time. 2) We present a dedicated method section of the analytical setup that has been developed at MPI-BGC to analyse $\delta^{13}$C-CH$_4$ and $\delta^2$H-CH$_4$ in air samples. 3) We present a dedicated method section on the origin/development of the scale anchors at IMAU and MPI-BGC. 4) We have improved and clarified the applied terminology. For example we define the use of "calibration" in comparison to "measurement", we refer to the produced gas mixtures as synthetic CH$_4$-in-air standards and to the pure CH$_4$ gases as primary and secondary CH$_4$ gases. "Working standard" is abbreviated WS, "certified reference material" CRM, "reference material" RM and "matrix reference material" (e.g. CH$_4$ in air) as m-RM, complying with recommendations from IAEA TecDoc 1350. 5) We revised the calculation of the uncertainties and dedicated a separate method section to present our calculation method. All data presented in the manuscript include the uncertainties of the full traceability chain where possible (CRM➔WS➔primary CH$_4$➔secondary CH$_4$). The presented uncertainties include the most recent development in CRM's, i.e. the new uncertainty for LSVEC. 6) We revised the comparison between MPI-BGC and the previously published data/method of Sperlich et al., (2012). This includes a revision of the Sperlich et al., (2012) data and their uncertainties to include the full traceability chain. Moreover, we present new comparison experiments between the two methods of MPI-BGC and Sperlich et al., (2012) to discuss/support the methods presented in this manuscript. 7) The experiments to evaluate the potential for analytical errors of the new methods are explained and discussed in the main text in more detail, full details are provided in the Appendix.

Our response to the reviewers comments is indicated in blue in the following.

**Reviewer 2 (Ingeborg Levin)**

**General Comments:**

First, I want to congratulate Sperlich and co-workers for their extremely valuable and important work, which will hopefully, more than three decades after the first publications of isotope measurements in atmospheric methane, solve our problem of lack of reference standards for these analyses. Sperlich et al. present a sound way for linking carbon and hydrogen isotope ratios in pure $CH_4$ to the internationally accepted IAEA carbonate and water reference materials. After dilution of these calibrated pure $CH_4$ gases with $CH_4$-free synthetic air they produce $CH_4$-in-air mixtures of ambient concentrations that can be used in the future as calibration standards, linking atmospheric (and source) methane isotope analyses from globally distributed labs to a common calibration scale. As was already pointed out by Referee # 1, this fundamental work will become one of our basic references to describe the development of our future methane isotope calibration scale. As such, however, the descriptions of procedures in the current version of the manuscript, unfortunately, do not fully meet the requirements for clarity and completeness. Referee # 1 has already prepared a long list of comments and made very good suggestions for improvements of the manuscript, which I fully support. In my list of comments below, I thus only want to re-emphasize a number of points, which I feel most important to be tackled in a revised manuscript.

**Specific Comments:**

1. Introducing the various standard materials, their production (e.g. also by spiking with deuterated $CH_4$), hierarchy and their calibration against IAEA reference materials (rm), or against other $CH_4$ gases or other $CH_4$-in-air gas mixtures is rather confusing. This does not only concern Figure 1 and Tables 1 and 2, but also Table 3, where the calibration results are given. I like very much the revised Figure 1 suggested by Referee #1.
OK, a similar figure has been added to the revised manuscript. We hope this clarifies the hierarchies and relations between applied gases and reference materials.

Please also be VERY clear with your nomenclature, e.g. distinguishing between "calibrations" (i.e. against reference materials) and what, from my point of view should better be named "comparison" with the earlier MPI-Jena standard gases Carina.
OK. We define our use of calibration and measurement at the beginning of the method section.

In fact, it is not really clear to me how the $H_2$ scale in Brass and Röckmann (2010), which forms the basis of the earlier MPI-Jena scale has been established. In their paper Brass and Röckmann refer to a paper by Bergamaschi et al. (1994) who obtained their calibration from colleagues at BGR, Hannover.
We dedicate a new section on the description of the scale history at IMAU.

The observed $\delta^2H$ difference of 4‰ between the IMAU/earlier MPI-BGC scale and the recent calibration may perhaps not be surprising.

That is true, it is not surprising. It is rather surprisingly good considering the previously achievable measurement precisions and scale propagations. We think this is better discussed in the revised manuscript.

What does the remark on page 12 line 24-25 mean in this context? More information about the origin of the IMAU/earlier MPI-BGC scale is required to judge on the comparison results listed in Table 4.

OK, detailed information are provided in revised manuscript.

I think, the sentence in the conclusion P17, L 27 is too strong as the earlier MPI-BGC scale is only propagated from some yet unexplained origin.

OK, section is re-formulated and considers the concerns on propagated scale anchor.

2. Concerning the experimental set up for the calibrations against carbonates and water, I find the descriptions confusing and much too brief. A figure that displays the complete setups (for $CH_4$ against carbonates and for $CH_4$ against water) would be very helpful. Figures 2 and 4 could then be integrated there.

New figures are included in the manuscript that show the reactors of the TC/EA-IRMS and the EA-IRMS system. Both figures contain the 10-port valve configuration to display the $CH_4$ injection into each of the systems. All other reference materials are introduced via autosampler in both systems. We think the descriptions in the revised manuscript are more clear and detailed.

3. Discussion on accuracy of the calibrations: Although the authors have explained in detail how they tried to follow, as much as possible, the principle of identical treatment (PIT) and to avoid possible pitfalls when calibrating $CH_4$ against carbonate and $H_2O$ reference materials, they cannot be sure that indeed no systematic biases have occurred. The most convincing argument for accuracy of the new standards to me is the good agreement with the earlier work by Sperlich et al. (2012) who used a (slightly) different procedure than in the present work. The discussion of the uncertainty in this respect is not clear enough. It seems rather to come as a mixture of long-term precision, agreement with the IMAU scale (see my reservations above) and finally arguing with "the combined uncertainty". I would like to see here a more elaborated discussion and clear separation of the different indicators for accuracy. This could hopefully help to pin down biases in the future.

OK. The revised manuscript has a strong focus on the comparison with the earlier work of Sperlich et al., (2012). In fact, we present results from new experiments to compare the methods from CIC and MPI-BGC in more detail. In order to do so, we revised the data evaluation and the uncertainty calculation from Sperlich et al., (2012). We agree this part has fallen short in the previous version of the manuscript and think it is much clearer in the revised version. While we address the differences in the comparison, our explanation for the cause of the differences is limited. We discuss the potential for measurement artefacts, such as incomplete conversion or scale compression effects. However, we discuss why we think that we have good control on these processes and that we can therefore not identify the cause of the inter-laboratory differences.

4. I also agree with Referee #1 that a description (or at least a reference to a publication) of the iSAAC measurement system is required.
OK, included. As mentioned above, the respective paper has just been published (Brand et al., 2016).

**Minor comments:**

The Appendix is named Appendix 1 in the text but A and B when they show up
OK.

Sect. 3.2: that the high $\delta^2H$ values have been produced by spiking should go into section 2.1
OK.

In the discussion section it may be helpful to explain why $m/z$ = 15 is used to detect unconverted $CH_4$ in the sample.
OK, explanation included in revised manuscript.

---

## Author Comment (AC3) · 10 Jun 2016

**Author's Comments to the reviewers**

First of all, we would like to thank both reviewers and the author of the additional comment for their insightful and constructive critiques. As suggested by both reviewers, we have considerably re-written and re-structured large parts of the manuscript. Therefore, page and line numbers of the specific comments by the reviewers will not match the numbers in the revised manuscript. The revised manuscript considers every point mentioned by the reviewers and the comment, unless specifically addressed otherwise. We dedicated specific emphasis on the following aspects:

1) In the introduction, we present a review on the history of produced scale anchors at a range of laboratories analysing $\delta^{13}$C-CH$_4$ and $\delta^2$H-CH$_4$. This list provides a good overview on the magnitude of potential inter-laboratory offsets and their variability over time. 2) We present a dedicated method section of the analytical setup that has been developed at MPI-BGC to analyse $\delta^{13}$C-CH$_4$ and $\delta^2$H-CH$_4$ in air samples. 3) We present a dedicated method section on the origin/development of the scale anchors at IMAU and MPI-BGC. 4) We have improved and clarified the applied terminology. For example we define the use of "calibration" in comparison to "measurement", we refer to the produced gas mixtures as synthetic CH$_4$-in-air standards and to the pure CH$_4$ gases as primary and secondary CH$_4$ gases. "Working standard" is abbreviated WS, "certified reference material" CRM, "reference material" RM and "matrix reference material" (e.g. CH$_4$ in air) as m-RM, complying with recommendations from IAEA TecDoc 1350. 5) We revised the calculation of the uncertainties and dedicated a separate method section to present our calculation method. All data presented in the manuscript include the uncertainties of the full traceability chain where possible (CRM➔WS➔primary CH$_4$➔secondary CH$_4$). The presented uncertainties include the most recent development in CRM's, i.e. the new uncertainty for LSVEC. 6) We revised the comparison between MPI-BGC and the previously published data/method of Sperlich et al., (2012). This includes a revision of the Sperlich et al., (2012) data and their uncertainties to include the full traceability chain. Moreover, we present new comparison experiments between the two methods of MPI-BGC and Sperlich et al., (2012) to discuss/support the methods presented in this manuscript. 7) The experiments to evaluate the potential for analytical errors of the new methods are explained and discussed in the main text in more detail, full details are provided in the Appendix.

Our response to the reviewers comments is indicated in blue in the following.

**Comment 1 (Sergey Assonov)**

Stable isotope measurements of greenhouse gases $CO_2$ and $CH_4$ make a powerful tool used to understand processes involved in the global carbon cycle. In order to get meaningful interpretation of stable isotope data in greenhouse gases, the data produced in different labs and in different years should be compatible within certain limits (WMO, 2014), these are called as compatibility goals (Table 1). In the last years compatibility of air-$CO_2$ stable isotope data is thought to be improved by introducing calibration "JRAS air" mixtures (WMO, 2012).

The compatibility goals for $CH_4$ are still a challenge to be achieved; it can be realized by using optimised calibration schemes, and to be based on appropriate reference materials with low uncertainty. All in all the compatibility goals can be considered in the first instance as the uncertainty required for "fit-for purpose" reference materials. In turn, such reference materials have to be compatible with a sample, that is why the community needs several reference $CH_4$-in-air mixtures.

| Table 1. Compatibility goals for atmospheric $CH_4$ (after WMO, 2014). Component | Compatibility goal (for background air) | Extended compatibility goal (for polluted air) |
|---|---|---|
| δ13C-CH4 | ± 0.02‰ | ± 0.2‰ |
| δ2H-CH4 | ± 1‰ | ± 5‰ |

Thereafter, work on the calibration of pure $CH_4$ gases aimed to produce reference $CH_4$-mixures cannot be published without thoughtful considerations of the uncertainty estimation and without clear presentation of the uncertainty budget. In particular this is expected for the work presented by the WMO-GAW Central Calibration Lab for stable isotopes in greenhouse gases (currently MPI-BGC, Jena, DE). In this respect the manuscript demonstrates serious problems such as unclear presentation of the calibrating approach in general, unclear uncertainty budget, as well as potentially missing/neglecting some uncertainty components. The major shortcuts are as following:

1. In order to build a skeleton of the uncertainty propagation, one has to consider a traceability chain for all measurements. The traceability chain has to be tracked to the highest Ref. Materials (RMs) in use. In case of $\delta^{13}C$ these are NBS19 & LSVEC (these have to be considered with their uncertainties) and include all measurement steps. Each next measurement step (including measurements on RMs) introduces an analytical uncertainty, thus increasing the total uncertainty.

OK, that is included in Section 2.4 of the revised manuscript. Please keep in mind that per definition, NBS 19 has no uncertainty, as did LSVEC until just before the first version of the manuscript was submitted. The revised manuscript includes the calculation of the full traceability chain. In that, we calculated two uncertainties, with and without the new uncertainty of LSVEC.

2. The uncertainty propagation should be based on the traceability chain and also include all potential effects due to TC/EA, gas dilution etc. Besides, I would suggest to present the uncertainty budget, namely to describe a contribution of each uncertainty component starting from the uncertainty assigned to RMs carbonates (NBS19 and LSVEC), then the uncertainty of carbonate measurements, the uncertainty of 2-point calibration as based on the carbonates, then analytical uncertainty of "master"-$CH_4$ (this is used for calibration "calibration"$CH_4$) etc. Such uncertainty budget will clearly demonstrate where further improvements are essential.

OK, see above comment. The uncertainty budget is revised and includes the uncertainties of all hierarchy levels. We explain in detail what systematic errors we suspected and how we were able to exclude them.

3. The "master" $CH_4$ (and its replacement when the first "master" was lost) was calibrated vs the IAEA Ref. Materials by applying the 2-point calibration approach. Next, several "calibration" $CH_4$ were calibrated vs the "master" $CH_4$. It is unclear how the 2-point calibration was applied in the case of measuring several "calibration" $CH_4$ gases? In fact calibration vs. the "master" $CH_4$ looks like 1-point, thus violating the 2-point calibration approach (Coplen et al., 2006) designed to address various effects during sample preparation and measurements. I stress – this is in particular critical for $\delta^{13}C$ values being down to -69.9 ‰ (Tab 3 in the manuscript), far below -40 ‰ of the "master"$CH_4$ and also outside the LSVEC value of -46.6 ‰.

OK, the revised manuscript considers this point in detail. We mention when scale compression corrections have been applied and present a comparison of isotopic differences between two very different gases as determined with each of the different methods. The agreement in scale resolution between the applied method is excellent. We also address the need for the replacement of LSVEC to extend to $\delta^{13}C$ range found in biogenic $CH_4$ in order to tackle this problem in future.

4. Given that "calibration" $CH_4$ gases were characterised against the "master" $CH_4$, it is unclear why the $\delta^{13}C$-uncertaitnty of 0.06 ‰ for Martha-1 ("calibration"-$CH_4$) is smaller than the uncertainty of 0.07 ‰ obtained for the "master" $CH_4$. The uncertainty of each next material cannot be smaller than the uncertainty of material(s) used for its calibration (in this case uncertainty of "master" $CH_4$). This example implies something to be wrong in the uncertainty evaluation scheme in general. For the same reasons the $\delta^{13}C$ uncertainty of ± 0.08 ‰ given for the "calibration" $CH_4$ Mike-1 looks like optimistically too low.

OK, the uncertainty calculations are revised (and version checked).

5. The authors should also explain the uncertainty values for "Biogenic" and "Fossil" $CH_4$, namely the values of ± 0.04‰, as given with the reference to (Sperlich et al., 2012). Why these are lower than uncertainties obtained by the work presented in this manuscript? In fact Sperlich et al. (2012) gave no detailed explanation on the uncertainty propagation. Given that the paper by Sperlich et al. (2012) is written by the same authors as the present manuscript, this is a must-requirement.

OK, we discuss the uncertainty of Sperlich et al., (2012) and revise the uncertainty estimate to include the full traceability chain.

6. When focusing high accuracy values, the authors need to consider the effect $^{17}O$ correction for the entire $\delta^{13}C$-calibration scheme, namely when calibration started from carbonates is applied to $CH_4$ gases. Is there any potential bias?

OK, $^{17}O$ correction and impact on $CH_4$ and carbonate analyses is addressed.

7. Last but not least, the authors wrongly cite the $\delta^{13}C$-uncertainty of LSVEC. The message sent in Dec-2016 by the IAEA to LSVEC customers suggests the $\delta^{13}C$-uncertainty of LSVEC at ±0.15 ‰; this value is also used by A. Schimmelmann et al. 2016 (see http://pubs.acs.org/doi/abs/10.1021/acs.analchem.5b04392). The present interpretation of the message distributed by the IAEA is misleading.

We adopted the uncertainty of 0.15 ‰ for LSVEC in the revised version of the manuscript.

All in all I find the uncertainty evaluation presented in the manuscript as unclear, confusing and partly misleading.

OK, this is re-visited in the new version of the manuscript.

The uncertainty evaluation for $\delta^2H$ may suffer for similar reasons.

Speculation.

Given the problem with LSVEC, the $\delta^{13}C$ uncertainty presently achieved appears not fulfilling the requirements.

OK, considered in revised manuscript.

Sergey ASSONOV (reference material specialist for stable isotopes)
IAEA Environment Laboratories, IAEA